# Two Sides of the Coin: Mast Cells as a Key Regulator of Allergy and Acute/Chronic Inflammation

**DOI:** 10.3390/cells10071615

**Published:** 2021-06-28

**Authors:** Zhongwei Zhang, Yosuke Kurashima

**Affiliations:** 1Department of Innovative Medicine, Graduate School of Medicine, Chiba University, Chiba 260-8670, Japan; zhangzwei22@gmail.com; 2Department of Mucosal Immunology, The University of Tokyo Distinguished Professor Unit, The Institute of Medical Science, The University of Tokyo, Tokyo 108-8639, Japan; 3International Research and Development Center for Mucosal Vaccines, The Institute of Medical Science, The University of Tokyo, Tokyo 108-8639, Japan; 4CU-UCSD Center for Mucosal Immunology, Department of Pathology/Medicine, Allergy and Vaccines, University of California, San Diego, CA 92093-0063, USA; 5Mucosal Immunology and Allergy Therapeutics, Institute for Global Prominent Research, Graduate School of Medicine, Chiba University, Chiba 260-8670, Japan

**Keywords:** mast cells, allergy, inflammatory bowel disease, immunotherapy

## Abstract

It is well known that mast cells (MCs) initiate type I allergic reactions and inflammation in a quick response to the various stimulants, including—but not limited to—allergens, pathogen-associated molecular patterns (PAMPs), and damage-associated molecular patterns (DAMPs). MCs highly express receptors of these ligands and proteases (e.g., tryptase, chymase) and cytokines (TNF), and other granular components (e.g., histamine and serotonin) and aggravate the allergic reaction and inflammation. On the other hand, accumulated evidence has revealed that MCs also possess immune-regulatory functions, suppressing chronic inflammation and allergic reactions on some occasions. IL-2 and IL-10 released from MCs inhibit excessive immune responses. Recently, it has been revealed that allergen immunotherapy modulates the function of MCs from their allergic function to their regulatory function to suppress allergic reactions. This evidence suggests the possibility that manipulation of MCs functions will result in a novel approach to the treatment of various MCs-mediated diseases.

## 1. Introduction

Mast cells (MCs) are a type of innate immune cell that belongs to the myeloid lineage. It is generally believed that MCs from both humans and rodents are derived from hematopoietic stem cells (HSCs). Mast cell progenitors (MCPs) leave the bone marrow as immature cells and enter the blood circulation, with the help of surface molecules, such as α4β7 integrin, MAdCAM-1 and VCAM1, to migrate to various target tissues [1,2]. MCs have high plasticity and heterogeneity due to their unique process of maturation. MCs are divided into two subsets in mice: mucosal mast cells (MMCs) and connective tissue mast cells (CTMCs) [3]. Similarly, in humans, MCs subtypes are classified according to whether they secrete both tryptase and chymase (MCTC) or tryptase only (MCT). MCs can express various receptors, and there are two main pathways of MCs activation: IgE-dependent and IgE-independent [4]. MCs are also regulators of inflammatory disorders and fibrosis occurred in various organs. Associations between MCs recruitment/infiltration and fibrosis have been found in various tissues [5]. Current studies have found that many MCs products, including—but not limited to—tryptase, chymase, histamine, TGF- β1, IL-13, IL-9, CCL2, platelet-derived growth factor (PDGF), glycosaminoglycan and fibroblast growth factor-2 (FGF-2) can promote fibrosis [5]. MCs have long been regarded as the initiators of allergy and inflammation, as well as the promoters of fibrotic diseases, which are pathogenic. However, as every coin has two sides, in addition to some harmful effects, MCs possess anti-allergic and inflammatory effects. MCs appear to play an immunomodulatory role in allergic, acute, and chronic inflammation (e.g., fibrosis [6,7]). There is growing evidence that MCs play an enormous role in allergic responses, inflammatory responses, and wound healing [6,8,9]. In view of the high heterogeneity and plasticity of MCs, MCs-mediated immune diseases are very complex, so many diseases have not yet been properly treated (e.g., food allergy [10,11]). Accordingly, MCs may become a new “drug target” if their unique characteristics can be targeted.

## 2. Origin and Heterogeneity of MCs

MCs are innate immune cells of the myeloid lineage, firstly named by Paul Ehrlich in 1878 according to their granular histological staining with aniline dye [3]. It is generally believed that MCs from both humans and rodents are derived from HSCs. In the bone marrow, HSCs first grow into myeloid progenitors and then differentiate into MCPs [3]. Next, MCPs transfer from the bone marrow as immature cells into the bloodstream. Eventually, with the help of surface molecules (e.g., α4β7 integrin, MAdCAM-1, and VCAM1) [2], MCPs migrate to various target tissues, such as the serous cavity, in close contact with the micro-environment (e.g., the skin, gastrointestinal tract, upper airways and lungs [12]) and some vascularized organs (e.g., the liver and kidneys) [13,14]. These tissues contain various tissue-specific factors (e.g., cytokines, growth factors and extracellular matrix [ECM]), which help MCPs finally become phenotypically mature and perform different functions [15].

However, the exact origin of MCs remains a matter of debate. It is generally believed that MCPs in mice are derived from bone marrow. It has also been reported that MCs are developed from the common myeloid progenitor cells (CMPs) [16]. However, Dahlin et al., who demonstrated that MCPs were derived from multipotential progenitor cells (MMPs) rather than CMPs, disagreed [17]. Recent studies have found that mouse MCs have dual developmental origins. A portion of the MCs have a primitive origin. They are derived from the yolk sac during embryogenesis. MCPs derived from the yolk sac migrate to different connective tissues, such as the skin. Another part of adult definitive MCs comes from the HSCs of the aortic-gonad-mesonephros vascular endothelium [18]. Studies have also shown that MCs in adult tissues are supplemented by the proliferation/differentiation of resident precursors in long-lived tissues [19]. Thus, in mouse-based studies, MCs have at least two maturation pathways; however, whether these pathways apply to other mammals, such as humans, has not been fully clarified.

MCs also have high plasticity and heterogeneity due to their unique process of maturation. They show subtype-dependent differences in cell morphology, histochemical characteristics, granular protease expression, function, and survival according to the microenvironment, activating factors and the cytokine milieu [20].

According to the traditional classification system, MCs are divided into two subsets in mice: MMCs and CTMCs. The two subsets have different anatomical localization and protease expression patterns [3]. CTMCs, as the name suggests, are mainly located in the connective tissue of the intestinal submucosa and muscularis propria, the peritoneal cavity, and skin. This subset expresses both chymase and tryptase [17]. Their cytoplasm also contains heparin proteoglycan and a high level of histamine [21]. In contrast, MMCs are a chymase-expressing type [17]. They are usually present in the mucosal tissues of the lung and gastrointestinal tract, with little or no heparin proteoglycans in their granules, and have lower amounts of histamine [21].

Similarly, in humans, the MCs subtypes are classified according to whether they secrete both tryptase and chymase (MCTC) or tryptase alone (MCT). The former is comparable with mouse MMCs, while the latter corresponds to CTMCs [20,21]. The MCTCs are mainly distributed in the skin, the gastrointestinal tract, and conjunctiva and the MCTs can be found in the lungs, nose and sinuses [22].

Nonetheless, as mentioned above, MCs show significant plasticity with different tissue microenvironments, activating factors and cytokine milieus [20]. Outside of the differences in protease content and anatomical localization, MCs also have heterogeneity in their receptor expression, release of mediators and responses to several stimuli. Hence, the traditional classification is too superficial. A consortium immunological genome project showed that MCs show distinct transcriptional differences from other major immune cells; however, there is also marked transcriptional heterogeneity between MCs recovered from different tissues in mice [23]. For example, in our previous study, we found that the extracellular ATP receptor P2X7, which is highly expressed in MCs of the colon, is lowly expressed in the skin [24,25]. Moreover, the MRGPRX2 receptor is highly expressed in MCs in the skin, while there is no sign of its expression in the lungs, which may be related to skin-specific effects, such as itchiness [26]. Furthermore, MCs can differentiate into distinct phenotypes, even within the same tissue, leading to more specific subpopulations [20,27]. In the human lung, for instance, MCs can express radically different levels of FcεR1 in different sites. MCs in alveoli and small-airways express lower levels of FcεR1 in comparison to those in the bronchi [28]. This may suggest that different locations in the same tissue also affect the phenotype and function of MCs. Likewise, MCs also have significant heterogeneity in the secretion of cytokines, chemokines and growth factors based on different species sources, tissue locations, developmental stages and exposure to inflammatory or immune responses, which was classified and elaborated in detail by Mukai et al. in a review [29].

Moreover, it has been found that the microorganism colonies in tissue are also involved in the development of MCs, which further complicates the heterogeneity of MCs. There is evidence that microorganisms in the skin can stimulate keratinocytes to release stem cell factor (SCF) in mice [30]. SCF is an essential factor for the proliferation, survival and differentiation of MCs in vivo; however, it is also a potent chemokine for MCs and circulating MCPs, which plays an indispensable role in the maturation of MCs [30].

Taken together, a new classification may be needed to accurately reflect the heterogeneity of MCs. Recently, a paper suggested that the traditional histological classification system for MCs should be replaced by a system based on the protease expressed by MCs, to more clearly show the tissue-specific versatility of MCs [31].

## 3. Pro-Allergic and Inflammatory Actions of MCs

### 3.1. Regulation of Receptor-Mediated MCs Activation

As one of the most common immune cells, the main characteristics of MCs are their pro-allergy and pro-inflammatory effects. MCs can express a variety of receptors, which, when combined with corresponding ligands, can induce the activation of MCs, thereby triggering various pathways. MCs activation mainly occurs by IgE-dependent and IgE-independent pathways [4].

MCs degranulation mediated by the IgE high-affinity receptor (FcεRI) is the most classical mechanism of MCs activation. Immunoglobulin E (IgE) is a class of antibodies produced by plasma cells that shows high affinity to MCs, and which can mediate Type I hypersensitivity reactions, such as food allergy and asthma. The first pathogenic step of IgE-mediated type I hypersensitivity reactions is sensitization (Figure 1A). When the body’s first contact with allergens, antigen-presenting cells like monocytes-macrophages and dendritic cells (DCs) present antigen information to T helper lymphocytes, causing them to secrete cytokines. B lymphocytes then transform into plasma cells and secrete various allergen-specific IgE in response to the cytokines derived from T helper lymphocytes [32]. Specific IgE can bind to FcεRI on the surface of MCs. FcεRI is a type of high-affinity receptor of IgE that exists in the form of a trimer or tetramer. FcεRI contains one α chain, two identical γ chains and one β chain which is sometimes missing. In humans, FcεRI can be expressed as both αβγ2 and αγ2. In rodents, however, FcεRI is only expressed in the form of αβγ2 [32,33]. The extracellular domain of the α chain can bind to the Fc segment of IgE, which is a critical site for triggering allergic reactions. The primary role of the β subunit is to enhance the tyrosine kinase activity and calcium influx and then to amplify the expression of FcεRI on the surface of MCs [17]. When the allergens enter again, FcεRI/IgE complexes are cross-linked with high-affinity antigens on the surface of sensitive MCs, the FcεRI receptor will be activated, causing signal transduction in MCs and promoting the degranulation of MCs and the subsequent release of inflammatory mediators, like histamine, serotonin and leukotriene, which are involved allergic reactions or inflammation [32].

The degranulation of MCs, which is mediated by IgE-FcεRI, mainly depends on the Lyn-Syk pathway (Figure 1A). In this pathway, the Lyn and Syk kinases are activated at first, and then phosphorylated Lyn and Syk activate some junction proteins (e.g., SLP-76, Gab2 and Vav1) with the help of the linker for activation of T cell (LAT), causing the downstream effector protein PLCγ1/2 to participate in the reaction [34]. This will facilitate the hydrolysis of phosphatidylinositol diphosphate (PIP2) to inositol triphosphate (IP3) and diacylglycerol (DAG), thus initiating the PLC-IP cascade, resulting in calcium mobilization and the activation of protein kinase C (PKC). Finally, the degranulation of MCs is mediated [35]. It is worth mentioning that another protein on MCs, non-T cell activation linker (NTAL), has a competitive inhibition with LAT. It can reduce the phosphorylation of LAT by activated-Syk through the competition as kinase substrates, thus producing negative regulation on FcεRI-mediated signaling pathway [36]. Additionally, phosphorylated Lyn can activate Btk and Emt, which are the keys to calcium mobilization and PKC activation [37]. At the same time, activated Btk can also activate the downstream NF-κB, MAPK and PI3K/Akt pathways [38,39]. Likewise, B cell lymphoma 10 (Bcl10) and mucosa-associated lymphoid tissue 1 (Malt1) were found to play an important role in the activation of NF-κB, which helps the production of pro-inflammatory cytokines such as TNF-α and IL-6 (Figure 1A) [40].

Furthermore, Fyn also plays a pivotal part in the degranulation of MCs (Figure 1A). Fyn is another non-receptor tyrosine-protein kinase of the Src family, which is located on the cell membrane. When FcεRI is activated, Fyn activates phosphatidylinositol-3-kinase (PI3K) through interaction with Gab2, thereby affecting calcium mobilization and regulating the degranulation of MCs [41]. Meanwhile, Gab2 involves the activation of downstream PI3K-AKT and other transduction pathways. PI3K and Rac-GTPases (from the Rho-GTPase family) can promote the remodeling of cell scaffolds, resulting in the dispersion of MCs and chemotaxis. It has been found that the activation of Rac1 and Rac2 GTPases was reduced in Fyn-deficient MCs. However, retroviral-mediated expression of Fyn, constitutively active forms of Rac2 GTPases or PI3K in Fyn-deficient MCs can help restore cell chemotaxis [42]. Therefore, Fyn kinase, as their unique upstream, has an important position. Adapter NTAL also participates in Fyn-mediated pathway. It has been shown that NTAL signals to the mouse MCs cytoskeleton via Rho-GTPase and plays a negative regulatory role in the chemotaxis [43,44]. It is worth noting that there has been controversy about the regulatory effect of NTAL on FcεRI in MCs from different sources [36]. NTAL has been reported to be a negative regulator of FcεRI signaling in mouse MCs, whereas the opposite is true in humans or rats. In light of this, a research based on functional studies of BMMCs with NTAL knockdown and the corresponding controls confirmed that NTAL is a negative regulator of FcεRI-mediated pathways [36]. Experiments also showed that MCs without Fyn could not secrete inflammation-related leukotriene B4 and C4, cytokines IL-6, tumor necrosis factor (TNF), chemokine CCL2 (MCP-1) and CCL4 (MIP-1β) [45].

MCs can also be activated through IgE-independent pathways. Cytokines, some complement fragments, neuropeptide substance P, inflammatory mediators, β-defensins and exogenous molecules (e.g., artemisinin, PAMP9-20, etc.) are able to activate MCs through specific G-protein-coupled receptors (GPCR).

Many cytokines can mediate the activation of MCs. Results show that IL-3 can recruit and tend to the expression of MCs in the nasal mucosa and reflect the severity of allergic rhinitis. IL-6 can regulate the expression of protease activated receptors (PARs) on the MCs membrane; thus, participating in allergic inflammation. IL-9 can promote the release of IL-2 by MCs, leading to the expansion of CD25^+^ type 2 innate lymphoid cells (ILC2), thus activating Th9 cells, and promoting lung inflammation in cystic fibrosis [46]. IL-12 can not only regulate the expression of PARs but also stimulate MCs to secrete IL-4 by activating the transduction pathways of protein kinases AKT and ERK [47,48]. IL-15 can stimulate MCs to secrete IL-4 through the STAT6 pathway and participate in the differentiation and maturation of Th2 cells [49]. IL-33 acts synergistically with IgE to enhance the activity of MCs [50]. IL-33 also signals MCs to produce Th2 cytokines (e.g., IL-5 and IL-13) [51], which are involved in skin inflammation, lower airway inflammation and asthma [52,53,54]. Nonetheless, IL-33 also plays a protective role in worm infection and wound healing [55,56]. TNF α-2β can inhibit the proliferation of MCs [57]. TGF-β1 promotes MCs to secrete pro-inflammatory cytokines and IL-13, which induces inflammation and fibrosis [58]. Eosinophil chemokine receptor 3 (CCR3) can promote the secretion of IL-8 by bronchial epithelial cells and affect the degranulation of MCs. SCF binds to the c-kit receptors on the membrane of MCs, which helps to activate MCs, making them produce various inflammatory transmitters and cytokines [59].

The complement system is involved in specific and non-specific immunity of the body and can regulate immunity and mediate the antimicrobial response. Some of the complement fragments can regulate the activity of MCs. C3a is an allergic toxin molecule produced after the activation of the complement system. It can activate MCs to release inflammatory mediators and is a crucial chemokine of MCs [60]. The C3a receptor (C3aR) is expressed in human MCs lines (HMC-1, LAD2), primary human MCs derived from differentiated CD34^+^ and skin MCs. C3a activates the Gi family of the G proteins through the phospholipase C (PLC) pathway, which induces Ca2^+^ mobilization in human MCs, resulting in degranulation and chemokine production. Besides, C3a can promote the degranulation of MCs by inducing ERK1/2 phosphorylation [61]. There is evidence that arachidonic acid stimulates MCs to produce platelet-activating factor and histamine by producing C3a, leading to shock [62]. C5a is another complement fragment that affects MCs. Likewise, C5a was found to induce the degranulation of MCs through PLC [63].

Mas-related G-protein-coupled receptors (MRGPRs) are a new class of G-protein- coupled receptors, that are specifically distributed on the surface of peripheral sensory neurons and MCs (Figure 1B). There are four subtypes of MRGPRs in humans, MRGPRX1~X4, while human MCs only express MRGPRX2 [64]. According to a preliminary analysis, MRGPRX2 binds to basal secretin (mostly cationic peptides), activates MCs through the non-IgE transduction pathway, which induces the degranulation of MCs and plays a pivotal role in host defense, pseudo-allergy, pruritus, neurogenic inflammation and pain [64,65,66]. In gastrointestinal diseases, the latest research suggests that MRGPRX2 is also an important link. Evidence has shown that MRGPRX2-mediated MCs activation only occurs in the inflamed tissues of ulcerative colitis (UC), which can be used to distinguish between inflamed and non-inflamed UC. Besides, a study of the Ashkenazi Jewish UC exome chip case-control cohort found that the serine allele on Asn62Ser of MRGPRX2 was involved in β-arrestin-mediated desensitization, which helped fight against UC [67]. Similarly, there is a gene in mice that is homologous to the human receptor MRGPRX2: MRGPRB2. Natural endogenous ligands of MRGPRB2/MRGPRX2 have been reported, such as neuropeptides (e.g., substance P, VIP), antimicrobial peptides (e.g., β-defensin, LL-37) and PAMP9-20. Other ligands, such as bacterial products (e.g., quorum-sensing molecules) and synthetic drugs (e.g., tetrahydroisoquinoline [THIQ] motif and artemisinin) have also been reported (Figure 1B) [68,69]. The MRGPRX2-dependent immune response has been found to be faster and shorter than the immune response mediated by IgE [22].

Last but not least, P2X7 is also an important receptor that activates MCs (Figure 1C) [70]. Our previous studies demonstrated that the expression of P2X7 is tissue-specific. There is no expression of P2X7 in the skin, but it is highly expressed on colonic MCs, leading to the initiation and exacerbation of intestinal inflammation [24,25]. P2X7 is a type of P2 purinoceptor, an ATP-gated ion channel that can specifically recognize ATP and which then participates in inflammation by mediating the release of IL-1β and IL-18 (Figure 1C) [25]. Moreover, it has also been reported that ATP-P2X7-mediated MCs can releases IL-33, which protects against Helicobacter pylori (HP) and intestinal worms [71]. Interestingly, in addition to the damaged or apoptotic tissue cells that can release ATP and which lead to P2X7-mediated MCs activation, MCs—when stimulated—can also release ATP themselves, which further enhances the activation of MCs through autocrine ATP release, thus aggravating inflammation and allergy [72]. Furthermore, extracellular ADP can also be converted back to ATP through extracellular enzymes such as ATP synthase and adenylate kinase 2 (AK2), which also lead to the further activation of MCs (Figure 1C) [25,73]. Notably, recent studies have shown that ecto-nucleotide pyrophosphatase-phosphodiesterase 3 (E-NPP3, also called CD203c) has a negative regulatory effect on this process. E-NPP3 can degrade ATP by hydrolyzing pyrophosphate and phosphodiester bonds. When ATP-dependent activation is initiated, E-NPP3 can be rapidly expressed in MCs, thereby hydrolyzing extracellular ATP to prevent overactivation of the MCs, which helps prevent chronic allergy and inflammation (Figure 1C) [72].

### 3.2. Fibrogenic Actions of MCs

Fibrosis is a pathological process caused by overgrowth, hardening, and excessive scarring, which can occur in almost all tissues [74]. No matter what type of tissue, there is a standard feature, fibroblast activation, that is significant in the remodeling of fibrous tissue. The expression of α-smooth muscle actin (α-SMA) in activated fibroblasts increases. Then the fibroblasts differentiate into fibrotic-phenotype myofibroblasts, which promotes wound healing and fibrosis through the secretion of large amounts of ECM components (e.g., collagen and fibronectin) and physical contraction [75].

In the process of fiber formation, fiber molecular subunits are first synthesized in fibroblasts. They transfer from the endoplasmic reticulum to the Golgi complex and finally secreted into the intercellular matrix. In the intercellular matrix, the propeptides of procollagen molecules are cleaved by specific proteolytic enzymes and transformed into tropocollagen. After that, the monomeric subunits are assembled and grow linearly and laterally to form the collagen macromolecular complexes that form the fiber, and finally, the fiber is formed [76].

Although the exact relationship between inflammation and fibrosis is not clear at present, the relationship between MCs recruitment/infiltration and fibrosis has been reported in the intestines, lungs, kidneys, liver, heart and other organs [15,75,77,78,79,80]. In addition, toluidine blue staining showed that MCs was mostly adjacent to fibroblasts in animal skin simulating trauma and it has been found that a variety of cell surface proteins on fibroblasts can connect to the interaction between fibroblasts and MCs, such as membrane-bound stem cell factor, hyaluronic acid receptors, fibrinogen and gap-junctional intercellular communication (GJIC) [76,81]. Consequently, it is reasonable to assume that MCs are regulators of fibrosis in different organ systems. Current studies have found that many MCs products, including—but not limited to—tryptase, chymase, histamine, TGF- β1, IL-13, IL-9, CCL2, PDGF, glycosaminoglycan and FGF-2 can promote fibrosis [5]. These secretory components of MCs can not only promote fibroblasts to produce collagen but also participate in the extracellular stage of fiber formation [76].

Chymase is produced by the degranulation of MCs and is closely related to fibrosis. Chymase promotes fibroblast mitosis and promotes the synthesis and secretion of type I and III collagen in the ECM [82]. In both humans and mice, chymase can cleave the Phe8-His9 bond of the non-bioactive peptide angiotensin I (Ang I) to form its bioactive peptide angiotensin II (Ang II). Studies have shown that Ang II and its angiotensin receptor 1 (AT1) are involved in the process of liver fibrosis. They can promote the proliferation of hepatic stellate cells and induce them to express α-SMA and TGF-β1 and form type I collagen, which leads to liver fibrosis [83]. Chymase also activates endothelin-1, which is thought to promote organ fibrosis [84]. Furthermore, chymase itself can enzymatically cleave the precursors of pro-gelatinase B (MMP-9), TGF-β1 and c-propeptide from type I collagen molecules, which will enhance their activity [76,85].

Tryptase is another typical protease secreted by MCs, which can also promote fibrosis [75]. However, the mechanism is not the same as that of chymase. Evidence suggests that tryptase induces fibrosis by activating protease activator receptor-2 (PAR-2) [86]. PAR-2 has been shown to be expressed in fibroblasts and HSCs. The activation of PAR-2 can stimulate the proliferation of HSCs, collagen production and TGF-β1 production, which are important to fibrosis [87]. However, there is a great deal of disagreement on the signal pathway mediated by PAR-2 activation. A study of inflammatory bowel disease (IBD) found that tryptase promotes fibrosis by activating the PAR-2/Akt/mTOR pathway of fibroblasts [75]. Meanwhile, there are reports about the role of the PI3K/Akt/ROCK pathway in pulmonary fibrosis and the PAR-2 and PAR γ pathways in atrial fibrosis [88,89]. Furthermore, tryptase can not only activate the mitosis and increase the migration activity of fibroblasts but also make them produce type I collagen fibronectin and laminin, which promote the formation of fibrosis [76,90].

Histamine, another substance secreted by MCs, also induces fibrosis. It is reported that histamine stimulates fibroblast proliferation and collagen synthesis [91,92]. Histamine has four receptors: H1R, H2R, H3R and H4R. These all belong to the G-protein-coupled receptor family, and play different roles by coupling different G proteins to regulate different signaling enzymes or pathways [15]. A study on pulmonary fibrosis shows that the histamine receptor H2R is positively coupled with adenylate cyclase through the G_s_ protein, which stimulates human lung fibroblasts through cAMP/PKA-dependent signal transduction. This will significantly enhance the expression of α-SMA in human lung fibroblasts [93]. The role of histamine in the stimulation of fibrosis in human skin has also been reported [94]. Nonetheless, some studies have shown that histamine can inhibit the expression of α-SMA in human skin fibroblasts induced by TGF-β1 through the activation of the H1R receptor [95].

TGF-β1 is widely known to be one of the main driving factors of fibrosis. Its expression is significantly upregulated in fibrotic tissues [15]. Although MCs are not the only source of TGF-β1, this pleiotropic cytokine may be the most characteristic fibrogenic mediator produced by MCs [79]. TGF-β1 acts on fibroblasts, promoting their proliferation and migration and also inducing their differentiation into myofibroblasts. In addition, TGF-β1 can enhance fibronectin internalization [96]. MCs secrete TGF-β1, but TGF-β1 can also regulate them [97]. TGF-β1 can not only regulate the proliferation, migration/chemotaxis and apoptosis of MCs but also induce MCs to release more IL-13. IL-13 can directly stimulate the proliferation and differentiation of fibroblasts and simultaneously induce the production and activation of TGF-β1 in fibroblasts; this has a dual mechanism of promoting fibrosis [98,99]. It has also been reported that some other products of MCs, such as IL-9 [46] and CCL2, can interact with TGF-β1, resulting in fibrosis [100].

Glycosaminoglycans secreted by MCs assist fiber formation in the extracellular stage [76]. As mentioned above, in the ECM, the polymerization of tropocollagen macromolecules can be polymerized into microfibers, fibrils and fibers. Glycosaminoglycan can absorb water before that, which helps to increase the concentration at the time of the polymerization, thus facilitating the polymerization of the tropocollagen. Along with this, there is an electrostatic interaction between glycosaminoglycan and collagen. A recent study showed that heparin forms a bridge between two collagen molecules, which makes it possible to regulate the distance between them, thus determining the thickness of the fibrils [101]. Accordingly, MCs, as the only source of heparin and other glycosaminoglycan in tissues, have a great contribution to the formation of fibers. What’s more, granules released by MCs can also act as nucleators, which can be used as the starting molecular loci of collagen molecular polymerization [76].

Some studies have found that the enzyme released by MCs can digest the membrane-bound form of SCF to produce soluble SCF, which helps the maturation of MCs by stimulating KIT. It also promotes fibrosis [15]. Moreover, it has recently been found that chymase may have an inhibitory effect on fibrosis at the same time. Chymase can degrade two alarmins, IL-33 and HMGB1 [102], which have been shown to promote the progression of fibrosis. In addition, MCs can secrete matrix metalloproteinases (MMPs), an enzyme required for collagen fiber degradation [76]. Thus, MCs protease may have more than one role and needs to be treated rationally according to the development of the disease.

## 4. Regulatory-Type Actions of MCs in Allergy and Inflammation

MCs have long been regarded as the initiators of immunity and inflammation, which is pathogenic. For example, specific MCs proteases can promote inflammation, such as tryptase, chymase and carboxypeptidase A3 [103,104,105,106]. Tryptase was found to promote M1 macrophages polarization and inflammation via PAR2/FOXO1 pathway, which may be related to macrophage-associated inflammation in obese adipose tissue and atherosclerotic plaque [107]. Tryptase was also found to promote inflammation in osteoarthritis and skin [106,108]. Similarly, the pro-inflammatory effects of Chymase have been detected in several inflammatory diseases. Studies have shown that the pro-inflammatory effects of chymase may be related to the activation of several cytokines and growth factors, such as IL-1β, IL-6, IL-8, IL-18, TGF-β1, endothelin-1 and -2, and neutrophil-activating peptide 2 (NAP-2) [104]. In addition, the pro-inflammatory effects of carboxypeptidase A3 have also been mentioned in some cases of respiratory inflammation [109,110,111].

However, there are two sides to every coin. In addition to some harmful effects, the benefits of MCs to the human body cannot be ignored. MCs appear to play an immunomodulatory role in inflammation, allergy and fibrosis.

### 4.1. Regulation of Chronic Inflammation/Fibrosis/Wound Healing

Wound healing is a self-protective mechanism of tissues and organs. It includes three stages: inflammation, fibroblast migration/proliferation, and remodeling. Many results show that while the depletion of MCs cannot prevent tissue repair and remodeling, it can delay wound healing [9,112]. Hence, MCs can promote wound healing. It has been pointed out that MCs are involved in wound healing at several stages [9]. Under the influence of SCF released by keratinocytes as well as CCL2 (MCP-1) and IL-33, MCs gather at the edge of the wound in the first few days [113]. The TNF secreted by them can enhance the expression of XIIIa factor in dermal dendritic cells and then promote hemostasis and clot formation, which help reduce injury [114]. Through the secretion of histamine, lipid mediators, and VEGF, they increase vascular permeability and recruit other cells, such as neutrophils, to help heal the wound (Figure 2) [5]. In the proliferation stage of wound healing, the migration and proliferation of fibroblasts and the formation of collagen fibers are the key steps of wound healing [5]. MCs can release a variety of substances that interact with fibroblasts to promote wound healing (Figure 2). As mentioned before, proteases released by MCs can chemotaxis fibroblasts and promote their mitosis [76,90]. In addition, VEGF, IL-4 and basic fibroblast growth factor (bFGF) derived from MCs can stimulate the proliferation of fibroblasts [115]. Subsequently, fibroblasts synthesize fibers to aid wound healing. This review has mentioned that MCs play an essential role in promoting this step. Besides, MCs can secrete a variety of mediators including VEGF, IL-8, IL-4, nerve growth factor (NGF), FGF-2, PDGF and TGF-β1 to facilitate angiogenesis, fibrin production and re-epithelialization [17,114,116]. Vascular degeneration is the main physiological process in the remodeling stage, which results in the transformation of granulation tissue into collagen-rich scar avascular tissue [117]. Although there are some objections [118,119], the vast majority of studies support the involvement of MCs in the process of scar formation [94,120]. In addition to mediators, GJIC between MCs and fibroblasts or myofibroblasts in granulation tissue also acts as a bridge of cellular communication, mediating excessive fibrosis [81]. Notably, recent studies have found that zinc is also crucial in the process of wound healing. Zinc and MCs cooperatively activate GPR39-mediated fibroblasts through the PKC/MAPK/C/EBP pathway to induce the production of IL-6, which will finally promote wound healing [121].

The regulation of chronic inflammation by MCs has also been reported [6,7]. It is generally believed that CD4^+^CD25^+^Foxp3^+^ regulatory T cells (Tregs) are a crucial cell in graft-versus-host disease (GVHD), which can mediate immunosuppression, thereby inhibiting disease progression and significantly reducing the incidence and mortality of the disease [6]. Remarkably, MCs are also involved in this process. It was reportedly difficult for MCs-deficient mice to develop graft tolerance [6]; however, immune tolerance can be re-established after the infusion of bone marrow-derived MCs [122]. Further studies have shown that activated Tregs can release IL-9, a factor associated with the growth and activation of MCs, to induce the recruitment and activation of MCs, which mediates local immunosuppression and immune tolerance [6]. Moreover, activated MCs can release TGF-β, which further activates Tregs and regulate the release of IL-9 [123]. Besides, Tregs can stabilize the cell membrane of MCs through the OX40-OX40L receptor, thus inhibiting the degranulation of MCs mediated by FcεRI and reducing rejection (Figure 2) [124]. Meanwhile, other studies have found that MCs also block the development of GVHD without relying on Tregs [125]. Those studies showed that MCs can directly secrete negative immune regulatory factors, such as IL-10, to induce the production of tryptophan hydroxylase 1 (TPH1). TPH1 degrades tryptophan, and an environment with low tryptophan effectively inhibits T cell proliferation, thus halting the progression of the disease (Figure 2) [126,127]. Moreover, evidence shows that MCs can promote the recovery of the epithelial barrier and epithelial regeneration in IBD. At the same time, MCs can quickly respond to bacteria that cross the mucosal barrier and prevent the barrier dysfunction induced by inflammation. Furthermore, MCs also downregulate the production of pro-inflammatory cytokines in the early stages of chronic colitis [7,128]. These actions have extraordinary significance in inhibiting the process of chronic inflammation.

### 4.2. Regulation of Allergic and Inflammatory Diseases

There is growing evidence that MCs play an important role in inhibiting allergic reactions and inflammation. It has been found that oral immunotherapy (OIT)-induced desensitized MCs have a robust regulatory function and can cooperate with Tregs to form a regulatory network, which helps control food allergy [8]. It has been reported that the anti-allergic part of MCs is mainly realized through the production of regulatory cytokines, such as IL-2 and IL-10 [8]. Several studies have found that the inhibition of the immune responses in the skin, intestines and bladder by MCs depends on IL-10 [22]. MCs-derived IL-10 can induce the production of Tregs by DCs (Figure 2). However, how the immunosuppressive effect of IL-10 is regulated at the molecular level remains to be determined. Thymic-derived Tregs are a type of suppressive T cell, which accounts for 5–10% of circulating CD4^+^T cells [129]. Studies have shown that Tregs can reduce the production of allergen-specific IgE and pathogenic Th2 and inhibit the degranulation of MCs and basophils, which push forward an immense influence on the control of allergic symptoms [8,130]. Tregs impairment is related to a loss of tolerance, autoimmunity and allergy [130]. In a similar manner, IL-2 is also involved in MCs-mediated immunosuppression and is critical for the development, amplification, activity and survival of Tregs [131]. IL-2 secreted by MCs can effectively expand Tregs populations and then inhibit the immune response. Another cytokine, IL-33, can indirectly participate in the immunosuppressive response by stimulating MCs to secrete IL-2, promoting ST2-independent immunosuppression in Tregs (Figure 2) [132,133]. Meanwhile, some studies have shown that IL-33 can promote the secretion of IL-13 by MCs, which inhibits the production of IL-12 by DCs in the skin. This can hinder the Th1 cell response to cutaneous antigen exposure [57].

In inflammatory diseases, although the reason why the role of MCs changes from pro-inflammatory to anti-inflammatory has not been fully clarified, it is undeniable that MCs have an important role in the regulation of inflammation. A number of studies have shown that MCs can inhibit inflammation under the stimulation of IL-33. One of the studies found that MCs, induced by IL-33, can secrete IL-13, which stimulates the polarization of alternatively activated macrophages (AAMFs) (Figure 2). AAMFs, also known as M(IL-4) cells, are activated by the IL-4Ra signal [134]. The AAMF expression of arginase-1 can deplete the extracellular microenvironment of arginine, and arginine deficiency has an inhibitory effect on activated T cells [135]. Accordingly, AAMF has immunosuppressive and anti-inflammatory effects. Notably, IL-6, another cytokine secreted by MCs, can upregulate the IL-4ra in IL-13R, which further enhances the alternating activation of macrophages [136]. Besides, in a study of autoimmune diabetes, MCs were found to play a protective role by inducing Foxp3^+^Treg cells and reducing tissue inflammation [137]. Studies also show that programmed cell death-1 (PD-1) on the surface of MCs can inhibit effector CD8^+^ T cells through the PD-1/PD-L1 (PD-ligand 1) pathway, which plays a regulatory role in cutaneous allergic inflammation and tumors (Figure 2) [138].

## 5. Future Prospects in the Manipulation of the MCs Functions

### 5.1. Novel Prospective Mast Cell-Targeting Strategies

In view of the high heterogeneity and plasticity of MCs, MCs-mediated immune diseases are very complex, so many diseases have not yet been properly treated. Take food allergies, for example, there is no curative treatment for food allergies or feasible treatment for the prevention of allergic reactions currently. People usually try to control the disease by avoiding allergenic foods; however, accidents occur. The results of OIT are not satisfactory. Due to the frequency and/or severe adverse effects of the treatment, many patients discontinue OIT before it works [10,11]. Thus, finding a target for the treatment of autoimmune diseases has become a primary goal. As mentioned above, MCs, as a common immune cell, play a dual role, which means that although MCs cause allergies, they can also help treat allergic diseases if properly used. Accordingly, if their unique characteristics can be targeted, MCs may become a new “drug target”.

As mentioned in this review, food allergy is mediated by food-specific IgE, which binds to the high-affinity receptor FcεRI on the MCs membrane. When the receptor is cross-linked, MCs degranulate and release various mediators, leading to the production of leukotrienes and prostaglandins, which causes corresponding clinical symptoms [139]. Since FcεRI is responsible for all of the responses that involve IgE, this pathway holds promise for the prevention of food-induced anaphylaxis.

Omalizumab is an anti-IgE antibody that binds to the C3 domain of IgE, preventing free IgE from binding to FcεRI and reducing the sensitivity to allergens. However, Omalizumab is not effective for all patients, and further research is needed [140]. Clinical trials of talizumab, an anti-IgE monoclonal antibody developed by Tanox, are currently underway to evaluate its efficacy in individuals with peanut allergy [141]. Ligelizumab is a humanized anti-IgE monoclonal antibody created by Novartis. In comparison to omalizumab, it can combine with free IgE more efficiently. Its efficacy in food allergy needs to be evaluated [142]. Quilizumab is a humanized fucosylated anti-IgE antibody produced by Genentech Company. The difference is that it does not bind to free/bound IgE. By binding to M1’, a domain specific to membrane-bound IgE in B cells, it causes the apoptosis of B cells that produce IgE, thus reducing the amount of IgE in circulation. Unfortunately, although it reduces IgE levels, it does not seem to reduce allergic reactions [143]. MEDI4212 is a high-affinity antibody against IgE that can bind to free/bound IgE simultaneously, depleting IgE-producing B cells. This antibody seems to work more efficiently than omalizumab in high-IgE-level patients (Table 1) [144].

Regarding kinases, as mentioned above, many kinases are involved in MCs activation and FcεRI signal transduction, including (but not limited to) Syk, Lyn, and Fyn. Most of these enzymes are potential targets for inhibiting the IgE-mediated activation of MCs and basophils. Fostamatinib (Tavalisse), a drug used to treat immune thrombocytopenic purpura, has recently been reported to inhibit Syk [145]. GSK2646264 is another Syk inhibitor, which effectively prevents MCs activation; however, at present, it has not entered clinical trials due to its potential toxicity [146]. EGFR inhibitors WZ3146 can inhibit the activation of MCs in vitro through non-targeted antagonism of Lyn and Fyn [147]. AZD7762 can also inhibit the activity of Lyn/Fyn, but it has been reported to have cardiotoxicity (Table 1) [148]. It is worth mentioning that Siglec-8 is the most promising target for the treatment of allergic diseases. It is only expressed in human MCs, basophils and eosinophils [149,150]. It has been found that the humanized non-fucosylated IgG1 antiSiglec-8 antibody lirentelimab (Allakos) is effective in the treatment of atopic keratoconjunctivitis, inert systemic mastocytosis, eosinophilic gastritis and duodenitis (Table 1) [151].

In addition to the IgE-FcεRI pathway, it was mentioned above that Tregs are the key to maintaining immune self-tolerance, and that they also regulate the immune response to exogenous Ags (including allergens). Furthermore, the proliferation of Tregs is activated by IL-2. It has been reported that low-dose IL-2 can induce the expansion of Tregs, which has great potential in the treatment of food allergy (Table 1) [130].

### 5.2. Current Problems and the Direction of Future Development

Due to ethical and other constraints, we usually use mice with MCs defects for research. Unfortunately, there is no perfect MCs defective mouse model at the present time. For instance, in Kit W-sh/W-sh mice, c-Kit mutations can affect lineages other than MCs. Although a better alternative is now available, we can use mice in which the MCs deficiency reflects more selective genetic changes. Given that these experiments are still based on C57BL/6 or BALB/c mice, we do not know if the MCs in these varieties function precisely as they do in humans, which strain of mice that provides MCs that are most similar to those of humans remains to be seen. More importantly, humans are genetically outbred, while MCs are highly heterogeneous and plastic. Thus, there may be some differences from individual to individual. Inevitably, the accuracy of the experimental results may be affected.

“Personalized medicine” and “precision medicine” are inevitable trends in the development of medicine in the future. Hence, it is necessary for us to recognize the molecular basis of disease accurately and to make the corresponding diagnosis and implement appropriate treatment measures. However, the more sensitive cells are to environmental conditions, the more complex it may be to choose beneficial pharmacological substances. MCs with high heterogeneity and plasticity are a typical example. They express hundreds of mediators and surface receptors according to their microenvironment and may play beneficial or non-beneficial roles. Consequently, deciphering the complexity of MCs phenotypes under specific pathophysiological conditions and the targeting of specific MCs is the direction of future research on immune diseases mediated by MCs.

## Figures and Tables

**Figure 1 cells-10-01615-f001:**
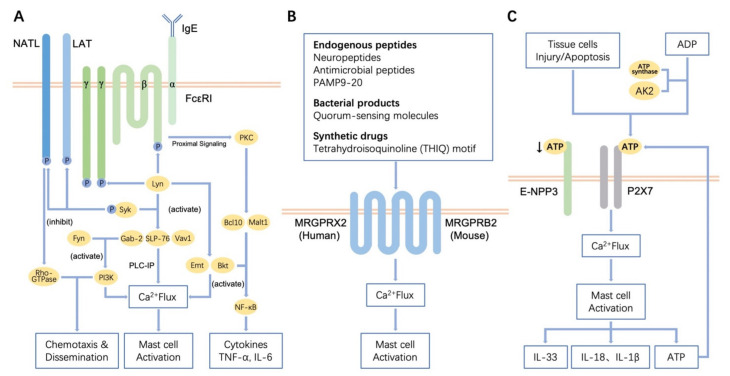
Regulation of receptor-mediated MCs activation. (**A**) IgE-FcεRI is the most classical mechanism of MCs activation, which mainly relies on the Lyn-Syk and Fyn pathways to activate MCs. (**B**) MRGPRX2/B2 is a novel class of G-protein-coupled receptors that can bind to various secretory proteins (mostly cationic peptides) to activate MCs. (**C**) P2X7 is an ATP-gated ion channel that is highly expressed in colonic MCs. P2X7 can also be activated by autocrine released ATP, and extracellular ATP can be degraded by E-NPP3 on the membrane.

**Figure 2 cells-10-01615-f002:**
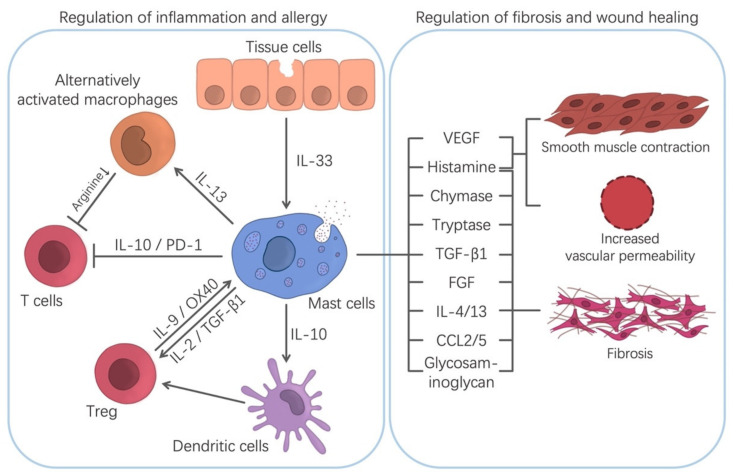
MCs in allergic reactions and inflammation. By releasing various cytokines and mediators, MCs can interact with other immune cells (e.g., they can promote the growth of Tregs and inhibit the generation of common T cells). Furthermore, MCs promotes smooth muscle contraction, increasing vascular permeability and promoting fibrosis. These processes suggest that MCs play an important role in wound healing, immune tolerance, and the suppression of allergies and inflammation.

**Table 1 cells-10-01615-t001:** Novel prospective MCs-targeting drugs.

Target	Name	Mechanism of Action	Characteristics	Reference
IgE-FcεRI	Omalizumab	Prevents free IgE from binding to FcεRI	Dose not effective for all patients	[140]
	Talizumab	Prevents free IgE from binding to FcεRI	Still in clinical trials	[141]
	Ligelizumab	Prevents free IgE from binding to FcεRI, may reduce B cell production of IgE	Combines with IgE more efficiently than Omalizumab	[142]
	Quilizumab	Depletes IgE-producing B cells	Does not seem to reduce allergic reactions	[143]
	MEDI4212	Prevents free/bound IgE from binding to FcεRI; depletes IgE-producing B cells	Probably better than omalizumab when IgE level is high	[144]
Kinases	Fostamatinib	Inhibits Syk	The effect is fast, but the safety is poor	[145]
	GSK2646264	Inhibits Syk	Has potential toxicity	[146]
	WZ3146	Inhibits Lyn/Fyn	Has no clinical data	[147]
	AZD7762	Inhibits Lyn/Fyn	Has cardiotoxicity	[148]
Siglec-8	Lirentelimab	Inhibits FcεRI signaling	The most promising target	[151]
Tregs	-	Induced by low doses of IL-2	Has great potential against food allergy	[130]

## Data Availability

No new data were created or analyzed in this study. Data sharing is not applicable to this manuscript.

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
