# Peer review of "Two Sides of the Coin: Mast Cells as a Key Regulator of Allergy and Acute/Chronic Inflammation"

_cells, 2021, doi:10.3390/cells10071615_

Round 1
Reviewer 1 Report
In this review authors present a general description of the activation mechanisms of mast cells and the reported roles of this cell type on distinct pathophysiological conditions. The participation of mast cells on inflammation, allergy, fibrosis and regulation of T cells is presented. Finally, authors mention the main existing strategies for the control of MC-dependent deleterious immune reactions. Although the review comprises main aspects of MC function and activation, in its present form, it fails to clearly present what is referred in the title as the “Two sides of the coin” of MCs actions. Besides, some important language problems were detected and some references are not correctly cited. Those deficiencies make difficult to understand the main message that authors want to transmit to the audience and do not contribute to elaborate a general conclusion.
Specific comments:
- In the introduction section it is not clear what are the "two sides of the coin". The existence of evidence of the role of MC in pro-inflammatory, anti-inflammatory and regulatory reactions is mentioned but it is not clear. Please re-phrase the introduction to make clear the reason why authors consider important the topic and the objective of the review.
- Line 18, please change protease by proteases
- Line 23, please change modulated by modulate
- Line 66, says that mast cells migrate from the yolk sac to different connective tissues, such as the skin, the most primitive origin. This sentence is not clear.
- Line 67 repeats two times the word “definitive”
- Line 98 must mention that there is a marked transcriptional heterogeneity between MCs recovered from different tissues in mice.
7. Line 118 mentions “Pro- allergic and inflammatory MCs”, this suggests the existence of a subtype of MCs that are pro-allergic and inflammatory. Since this has not been proven, Pro-allergic and inflammatory actions of MCs could be a more appropriate title.
8. Line 127 mention “Type I allergies”. This has to be replaced by Type I “hypersensitivity reactions”.
9. Line 132 states that “The cross-linking of specific IgE with FceRI on the surface of MCs will sensitize the MCs”. In general, the term “cross-linking” is applied when two igE receptors bound to respective monomeric IgEs are bringing closer by the action of an antigen or by high cytokinergic IgE. Crosslinking of the FceRI/IgE complexes with a high affinity antigen leads to degranulation, lipid mediator production and cytokine release. The term “sensitization” is used when monomeric IgE binds IgE receptors on MC surface and induce low-level of signaling leading to increase of MC survival and the synthesis of some chemokines and cytokines. Authors must adequate the sentence to accurately describe the IgE-dependent mechanism of activation of MCs.
10. Line 135. Authors should mention that, in humans, an isoform of the FceRI that lacks the beta chain is also expressed.
- Line 162 states “Fyn kinase, as their only upstream gene, has an important position”. Please clarify the meaning of the sentence.
- Line 164 to 165 Fyn-mediated pathway is associated with the phosphorylation of the adapter NTAL (Polakovicova, I., et al. PLoS One, 9(8):e105539, 2014), please include that information in the text and in Figure 1.
- Line 166 MIP-1 stands for Macrophage inflammatory protein 1 (chemokine CCL3), and is not related to carbon tetrachloride. Please correct.
Comment: FceRI signaling cascade activates NFkappaB transcription factor via PKCbeta, Bcl10 and Malt1 (Klemm, S., et al. JExpMed 203(2):337-347, 2006). This is relevant to the production of an important number of pro-inflammatory cytokines. That information must be included in the text and in Figure 1.
10. Line 167 “complements” refers to proteins of the complement system? Please clarify
11. Line 179 mentions the activation of an STAT26 patwhay activated by IL-15. However, a STAT pathway depending on a STAT26 protein has not been described so far (please correct). Besides, reference 63, that is cited mentioning that IL-15 can stimulate MCs to secrete IL-4 through “STAT26” pathway is a report of RAF kinase activation after IL-2 stimulation of T cells. The reference is misplaced. Please be sure that all the references in the text are correctly cited.
12. Line 253 mentions “Fibrogenic MCs”. This suggests the existence of a subtype of MCs that cause fibrosis. Since evidence on the participation of MCs on fibrosis is still controversial, and it has been related to the secretion of distinct mediators, the title should be changed to “Fibrogenic actions of MCs”
- Line 317 mentions “alarms”. Please substitute by “alarmins”
- Line 320 states “Regulatory-type MCs in allergy and inflammation”. As in other subtitles, the text better describes the regulatory-type actions of MCs in allergy and inflammation. With this in mind, the subtitle should be modified.
- Line 420 mentions the “dual role of mast cells” but, to this reviewer, that “dual role” has not been well delineated in the text. Do they refer to the pro-and anti-inflammatory roles of mast cells? If this is the case, why possible role in fibrosis and immune regulation is mentioned? Please clarify.
- Please mention a general conclusion about the “Two sides of the coin” on the role of MCs as key regulators of allergy and acute/chronic inflammation.
Author Response
To Reviewer 1
Comments and Suggestions for Authors
In this review authors present a general description of the activation mechanisms of mast cells and the reported roles of this cell type on distinct pathophysiological conditions. The participation of mast cells on inflammation, allergy, fibrosis and regulation of T cells is presented. Finally, authors mention the main existing strategies for the control of MC-dependent deleterious immune reactions. Although the review comprises main aspects of MC function and activation, in its present form, it fails to clearly present what is referred in the title as the “Two sides of the coin” of MCs actions. Besides, some important language problems were detected and some references are not correctly cited. Those deficiencies make difficult to understand the main message that authors want to transmit to the audience and do not contribute to elaborate a general conclusion.
- We appreciate to Reviewer#1 for pointing out the important points which need to be improved in the revised manuscript. We have answered all comments from reviewer and also precisely checked all references.
Specific comments:
- In the introduction section it is not clear what are the "two sides of the coin". The existence of evidence of the role of MC in pro-inflammatory, anti-inflammatory and regulatory reactions is mentioned but it is not clear. Please re-phrase the introduction to make clear the reason why authors consider important the topic and the objective of the review.
- We thank Reviewer#1 for pointing out the ambiguity in the review. “Two sides” refers to the beneficial and adverse effects of mast cells on the body. The introduction has been rewritten to clarify the “two sides” more clearly:
- MCs have long been regarded as the initiators of allergy and inflammation, as well as the promoters of fibrotic diseases, which are pathogenic. However, as every coin has two sides, in addition to some harmful effects, MCs possess anti-allergic and inflammatory effects. MCs appear to play an immunomodulatory role in allergic, acute, and chronic inflammation (e.g., fibrosis [6, 7]). (Page 1-2, line 45-50)
- Line 18, please change protease by proteases.
- We have changed “protease” by “proteases”, according to the reviewer's comment. (Page 1, line 18)
- Line 23, please change modulated by modulate
- We have changed “modulated” by “modulates”, according to the reviewer's comment. (Page 1, line 23)
- Line 66, says that mast cells migrate from the yolk sac to different connective tissues, such as the skin, the most primitive origin. This sentence is not clear.
- We thank Reviewer#1 for pointing out the ambiguity in the review. The sentence has been rewritten as:
- A portion of the MCs have a primitive origin. They are derived from the yolk sac during embryogenesis. MCPs derived from the yolk sac migrate to different connective tissues, such as the skin.” (Page 2, line 74-76)
- Line 67 repeats two times the word “definitive”
- We have removed the repeated “definitive” (Page 2, line 77)
- Line 98 must mention that there is a marked transcriptional heterogeneity between MCs recovered from different tissues in mice.
- We have added the word “in mice”. (Page 3, line 107)
- Line 118 mentions “Pro-allergic and inflammatory MCs”, this suggests the existence of a subtype of MCs that are pro-allergic and inflammatory. Since this has not been proven, Pro-allergic and inflammatory actions of MCs could be a more appropriate title.
- We thank Reviewer#1 for pointing out the error in the title. We have rewritten the title as “Pro-allergic and inflammatory actions of MCs” (Page 3, line 131)
- Line 127 mention “Type I allergies”. This has to be replaced by Type I “hypersensitivity reactions”.
- We have replaced “Type I allergies” by “Type I hypersensitivity reactions”, according to the reviewer's comment. (Page 3, line 140-141)
- Line 132 states that “The cross-linking of specific IgE with FceRI on the surface of MCs will sensitize the MCs”. In general, the term “cross-linking” is applied when two igE receptors bound to respective monomeric IgEs are bringing closer by the action of an antigen or by high cytokinergic IgE. Crosslinking of the FceRI/IgE complexes with a high affinity antigen leads to degranulation, lipid mediator production and cytokine release. The term “sensitization” is used when monomeric IgE binds IgE receptors on MC surface and induce low-level of signaling leading to increase of MC survival and the synthesis of some chemokines and cytokines. Authors must adequate the sentence to accurately describe the IgE-dependent mechanism of activation of MCs.
- We thank Reviewer#1 for pointing out sentences that describe inaccurately. We have rewritten the sentences as:
- The first pathogenic step of IgE-mediated type I hypersensitivity reactions is sensitization. When the body’s first contact with allergens, antigen-presenting cells like monocytes-macrophages and dendritic cells (DCs) present antigen information to T helper lymphocytes, causing them to secrete cytokines. B lymphocytes then transform into plasma cells and secrete various allergen-specific IgE in response to the cytokines derived from T helper lymphocytes [32]. Specific IgE can bind to FcεRI on the surface of MCs. FcεRI is a type of high-affinity receptor of IgE that exists in the form of a trimer or tetramer. FcεRI contains one α chain, two identical γ chains and one β chain which is sometimes missing. In humans, FcεRI can be expressed as both αβγ2 and αγ2. In rodents, however, FcεRI is only expressed in the form of αβγ2 [32, 33]. The extracellular domain of the α chain can bind to the Fc segment of IgE, which is a critical site for triggering allergic reactions. The primary role of the β subunit is to enhance the tyrosine kinase activity and calcium influx and then to amplify the expression of FcεRI on the surface of MCs [17]. When the allergens enter again, FcεRI/IgE complexes are cross-linked with high-affinity antigens on the surface of sensitive MCs, the FcεRI receptor will be activated, causing signal transduction in MCs and promoting the degranulation of MCs and the subsequent release of inflammatory mediators, like histamine, serotonin and leukotriene, which are involved allergic reactions or inflammation [32]. (Page 3-4, line 141-159)
- Line 135. Authors should mention that, in humans, an isoform of the FceRI that lacks the beta chain is also expressed.
- We thank Reviewer#1 for pointing out sentences that describe inaccurately. We have rewritten the sentences as:
FcεRI is a type of high-affinity receptor of IgE that exists in the form of a trimer or tetramer. FcεRI contains one α chain, two identical γ chains and one β chain which is sometimes missing. In humans, FcεRI can be expressed as both αβγ2 and αγ2. In rodents, however, FcεRI is only expressed in the form of αβγ2 [32, 33]. (Page 3-4, line 147-151)
We have added reference:
- [32] Pritchard, D.I.; Falcone, F.H.; Mitchell, P.D. The evolution of IgE-mediated type I hypersensitivity and its immunological value. Allergy 2021, 76, 1024-1040.
- [33] Cauvi, D.M.; Tian, X.; von Loehneysen, K.; Robertson, M.W. Transport of the IgE receptor alpha-chain is controlled by a multicomponent intracellular retention signal. J Biol Chem 2006, 281, 10448-10460.
- Line 162 states “Fyn kinase, as their only upstream gene, has an important position”. Please clarify the meaning of the sentence.
- We thank Reviewer#1 for pointing out sentences that describe inaccurately. We have rewritten the sentences as:
- It has been found that the activation of Rac1 and Rac2 GTPases was reduced in Fyn-deficient MCs. However, retroviral-mediated expression of Fyn, constitutively active forms of Rac2 GTPases or PI3K in Fyn-deficient MCs can help restore cell chemotaxis [42]. Therefore, Fyn kinase, as their unique upstream, has an important position. (Page 4, line 185-189)
We have added reference:
- [42] Samayawardhena, L.A.; Kapur, R.; Craig, A.W. Involvement of Fyn kinase in Kit and integrin-mediated Rac activation, cytoskeletal reorganization, and chemotaxis of mast cells. Blood 2007, 109, 3679-3686.
- Line 164 to 165 Fyn-mediated pathway is associated with the phosphorylation of the adapter NTAL (Polakovicova, I., et al. PLoS One, 9(8):e105539, 2014), please include that information in the text and in Figure 1.
- We thank Reviewer#1 to indicate the important points regarding the adapter NTAL. According to the reviewer's comment, we have added the description of NTAL as:
- It is worth mentioning that another protein on MCs, non-T cell activation linker (NTAL), has a competitive inhibition with LAT. It can reduce the phosphorylation of LAT by activated-Syk through the competition as kinase substrates, thus producing negative regulation on FcεRI-mediated signaling pathway [36]. (Page 4, line 168-171)
- Adapter NTAL also participates in Fyn-mediated pathway. It has been shown that NTAL signals to the mouse MCs cytoskeleton via Rho-GTPase and plays a negative regulatory role in the chemotaxis [43, 44]. It is worth noting that there has been controversy about the regulatory effect of NTAL on FcεRI in MCs from different sources [36]. NTAL has been reported to be a negative regulator of FcεRI signaling in mouse MCs, whereas the opposite is true in humans or rats. In light of this, a research based on functional studies of BMMCs with NTAL knockdown and the corresponding controls confirmed that NTAL is a negative regulator of FcεRI-mediated pathways [36]. (Page 4, line 189-196)
We have added reference:
- [36] Iva, P.; Lubica, D.; Michal, S.; Petr, D.; Cd., H.J. Multiple Regulatory Roles of the Mouse Transmembrane Adaptor Protein NTAL in Gene Transcription and Mast Cell Physiology. Plos One 2014, 9, e105539.
- [43] Tumova, M.; Koffer, A.; Simicek, M.; Draberova, L.; Draber, P. The transmembrane adaptor protein NTAL signals to mast cell cytoskeleton via the small GTPase Rho. Eur J Immunol 2010, 40, 3235-3245.
- [44] Halova, I.; Draberova, L.; Draber, P. Mast cell chemotaxis - chemoattractants and signaling pathways. Front Immunol 2012, 3, 119.
- Line 166 MIP-1 stands for Macrophage inflammatory protein 1 (chemokine CCL3), and is not related to carbon tetrachloride. Please correct.
- We thank Reviewer#1 for pointing out the error in the sentence. We have rewritten the sentence as:
- Experiments also showed that MCs without Fyn could not secrete inflammation-related leukotriene B4 and C4, cytokines IL-6, tumor necrosis factor (TNF), chemokine CCL2 (MCP-1) and CCL4 (MIP-1β) [45]. (Page 4, line 196-199)
- Comment: FceRI signaling cascade activates NFkappaB transcription factor via PKCbeta, Bcl10 and Malt1 (Klemm, S., et al. JExpMed 203(2):337-347, 2006). This is relevant to the production of an important number of pro-inflammatory cytokines. That information must be included in the text and in Figure 1.
- We thank Reviewer#1 to indicate the important points regarding the FceRI signaling. According to the reviewer's comment, we have added the description:
- Likewise, B cell lymphoma 10 (Bcl10) and mucosa-associated lymphoid tissue 1 (Malt1) were found to play an important role in the activation of NF-κB, which helps the production of pro-inflammatory cytokines such as TNF-α and IL-6 [40]. (Page 4, line 174-177)
We have added reference:
- [40] Klemm, S.; Gutermuth, J.; Hultner, L.; Sparwasser, T.; Behrendt, H.; Peschel, C.; Mak, T.W.; Jakob, T.; Ruland, J. The Bcl10-Malt1 complex segregates Fc epsilon RI-mediated nuclear factor kappa B activation and cytokine production from mast cell degranulation. J Exp Med 2006, 203, 337-347.
- Line 167 “complements” refers to proteins of the complement system? Please clarify
- We thank Reviewer#1 for pointing out sentences that describe inaccurately. “Complements” refers to C3a and C5a as mentioned in the review below. We have rewritten the sentences as:
- Cytokines, some complement fragments, neuropeptide substance P, inflammatory mediators, β-defensins and exogenous molecules (e.g., artemisinin, PAMP9-20, etc.) are able to activate MCs through specific G-protein-coupled receptors (GPCR). (Page 4, line 200-203)
- Line 179 mentions the activation of an STAT26 patwhay activated by IL-15. However, a STAT pathway depending on a STAT26 protein has not been described so far (please correct). Besides, reference 63, that is cited mentioning that IL-15 can stimulate MCs to secrete IL-4 through “STAT26” pathway is a report of RAF kinase activation after IL-2 stimulation of T cells. The reference is misplaced. Please be sure that all the references in the text are correctly cited.
- We thank Reviewer#1 for pointing out the error in the sentence and reference. We have rewritten the sentence as:
- IL-15 can stimulate MCs to secrete IL-4 through the STAT6 pathway and participate in the differentiation and maturation of Th2 cells [49]. (Page 5, line 211-213)
The reference is revised to:
- [49] Masuda, A.; Matsuguchi, T.; Yamaki, K.; Hayakawa, T.; Kubo, M.; LaRochelle, W.J.; Yoshikai, Y. Interleukin-15 induces rapid tyrosine phosphorylation of STAT6 and the expression of interleukin-4 in mouse mast cells. J Biol Chem 2000, 275, 29331-29337.
- Line 253 mentions “Fibrogenic MCs”. This suggests the existence of a subtype of MCs that cause fibrosis. Since evidence on the participation of MCs on fibrosis is still controversial, and it has been related to the secretion of distinct mediators, the title should be changed to “Fibrogenic actions of MCs”
- We thank Reviewer#1 for pointing out the error in the title. We have rewritten the title as “Fibrogenic actions of MCs” (Page 6, line 285)
- Line 317 mentions “alarms”. Please substitute by “alarmins”
- We have changed “alarms” by “alarmins”, according to the reviewer's comment. (Page 8, line 379)
- Line 320 states “Regulatory-type MCs in allergy and inflammation”. As in other subtitles, the text better describes the regulatory-type actions of MCs in allergy and inflammation. With this in mind, the subtitle should be modified.
- We thank Reviewer#1 for pointing out the error in the title. We have rewritten the title as “Regulatory-type actions of MCs in allergy and inflammation” (Page 8, line 384)
- Line 420 mentions the “dual role of mast cells” but, to this reviewer, that “dual role” has not been well delineated in the text. Do they refer to the pro-and anti-inflammatory roles of mast cells? If this is the case, why possible role in fibrosis and immune regulation is mentioned? Please clarify.
- We thank Reviewer#1 for pointing out sentences that describe inaccurately. We have added a supplementary explanation of the "dual role" in the review above:
- MCs have long been regarded as the initiators of allergy and inflammation, as well as the promoters of fibrotic diseases, which are pathogenic. However, as every coin has two sides, in addition to some harmful effects, MCs possess anti-allergic and inflammatory effects. MCs appear to play an immunomodulatory role in allergic, acute, and chronic inflammation (e.g., fibrosis [6, 7]). (Page 1-2, line 45-50)
- However, there are two sides to every coin. In addition to some harmful effects, the benefits of MCs to the human body cannot be ignored. MCs appear to play an immunomodulatory role in inflammation, allergy and fibrosis. (Page 8, line 398-400)
We have rewritten the sentences as:
- As mentioned above, MCs, as a common immune cell, play a dual role, which means that although MCs cause allergies, they can also help treat allergic diseases if properly used. (Page 11, line 511-513)
- Please mention a general conclusion about the “Two sides of the coin” on the role of MCs as key regulators of allergy and acute/chronic inflammation.
- We thank Reviewer#1 for pointing out the shortcomings of our expression. We propose that mast cells have "two sides", which have both pathogenic and therapeutic effects. In addition, MCs are highly heterogeneous and plastic, so the choose of beneficial pharmacological substances is complex. As a result, we conclude that “deciphering the complexity of MCs phenotypes under specific pathophysiological conditions and the targeting of specific MCs is the direction of future research on immune diseases mediated by MCs.” (Page 13, line 574-576)
Reviewer 2 Report
The article “Two sides of the coin: Mast cells as a key regulator of allergy and acute / chronic inflammation” by Zhongwei Zhang and Yosuke Kurashima discusses the role of mast cells in inflammation and allergy. The authors have done a lot of work to systematize the known data on the biology of mast cells in these aspects in general help for the reader to learn some new information about the problem under consideration. However, despite the significant number of references used, the authors were unable to disclose the role of mast cells in these aspects, taking into account modern data. Many references are outdated. There is a lot of much newer publications. This must be taken into account. In this regard, for the publication of the article, a significant revision of the work is necessary with the introduction of important and key references.
- Introduction
The current key works on the participation of mast cells in the pathogenesis of fibrosis are not cited.
- Origin and heterogeneity of MCs
It is necessary to consider the material of such key works as:
- Mast Cell Biology at Molecular Level: a Comprehensive Review. Elieh Ali Komi D, Wöhrl S, Bielory L. Clin Rev Allergy Immunol. 2020
- Mast cells as sources of cytokines, chemokines, and growth factors. Mukai K, Tsai M, Saito H, Galli SJ. Immunol Rev. 2018 Mar;282(1):121-150.
- Pro-allergic and inflammatory MCs – is an incorrect title of this section, since this type of mast cells is not included in the classification.
3.2. Fibrogenic MCs
What are Fibrogenic MCs? Can the author provide evidence of the existence of a MC of this type? The section name must be changed!
This section needs to be substantially supplemented and expanded.
This section focuses primarily on the mediated profibrotic effects of mast cells, albeit in insufficient detail. Some of the material is outdated works, and, despite their scientific value, should be supplemented by more recent literary sources. In addition, the authors do not consider at all the issues of the effects of the secretome of mast cells on the formation of collagen fibers, and even more so the direct effects on the molecular mechanisms of fibrillogenesis. At the same time, there are serious developments and studies on this matter, published, in particular, in the article:
- Atiakshin D, Buchwalow I, Tiemann M. Mast cells and collagen fibrillogenesis. Histochem Cell Biol. 2020 Jul; 154 (1): 21-40.
Data from this study should be included in the review.
- Regulatory-type MCs in allergy and inflammation
Regulatory-type MCs - it's not entirely clear what this is about. Maybe about a separate type of mast cells that have regulatory properties? Each mast cell has a wide regulatory spectrum in relation to immunocompetent cells, stromal cells, as well as fibrous and amorphous components of the extracellular matrix. In the opinion of the reviewer, the author should rename "Regulatory-type MCs" in the title of this section of the work; there is no clear evidence of the existence of this special type of mast cells.
4.1. Regulation of chronic inflammation / fibrosis / wound healing
Why do the authors in this section return to the fibrosis that was discussed earlier?
In addition, the authors cite data from rather old works on the participation of mast cells in regeneration and wound healing. Authors need to add materials from the following key works:
- A Review of the Contribution of Mast Cells in Wound Healing: Involved Molecular and Cellular Mechanisms. Komi DEA, Khomtchouk K, Santa Maria PL. Clin Rev Allergy Immunol. 2020 Jun; 58 (3): 298-312.
- A Review of the Evidence for and against a Role for Mast Cells in Cutaneous Scarring and Fibrosis. Wilgus TA, Ud-Din S, Bayat A. Int J Mol Sci. 2020
- Mast Cells in Skin Scarring: A Review of Animal and Human Research. Ud-Din S, Wilgus TA, Bayat A. Front Immunol. 2020 Sep 30; 11: 552205.
- Mast Cells in Diabetes and Diabetic Wound Healing. Dong J, Chen L, Zhang Y, Jayaswal N, Mezghani I, Zhang W, Veves A. Adv Ther. 2020 Nov; 37 (11): 4519-4537.
In addition, the authors in this section practically ignore the involvement of specific mast cell proteases in the development of inflammatory reactions. Therefore, the role of mast cells in inflammation is not adequately explained. It is necessary to add citations with data on the biological effects of tryptase, chymase, and carboxypeptidase A3, in particular:
- Protease profile of normal and neoplastic mast cells in the human bone marrow with special emphasis on systemic mastocytosis. Atiakshin D, Buchwalow I, Horny P, Tiemann M. Histochem Cell Biol. 2021 May; 155 (5): 561-580. "
- Mast cell chymase: morphofunctional characteristics. Atiakshin D, Buchwalow I, Tiemann M. Histochem Cell Biol. 2019 Oct; 152 (4): 253-269.
- Tryptase as a polyfunctional component of mast cells. Atiakshin D, Buchwalow I, Samoilova V, Tiemann M. Histochem Cell Biol. 2018 May; 149 (5): 461-477.
- Localization of Flt1 and tryptase of mast cells in skin wound of rats with type I diabetes: Initial studies. Bayat M, Chien S, Chehelcheraghi F. Acta Histochem. 2021.
Author Response
To Reviewer 2
Comments and Suggestions for Authors
The article “Two sides of the coin: Mast cells as a key regulator of allergy and acute / chronic inflammation” by Zhongwei Zhang and Yosuke Kurashima discusses the role of mast cells in inflammation and allergy. The authors have done a lot of work to systematize the known data on the biology of mast cells in these aspects in general help for the reader to learn some new information about the problem under consideration. However, despite the significant number of references used, the authors were unable to disclose the role of mast cells in these aspects, taking into account modern data. Many references are outdated. There is a lot of much newer publications. This must be taken into account. In this regard, for the publication of the article, a significant revision of the work is necessary with the introduction of important and key references.
- We appreciate to Reviewer#1 for pointing out the important points which need to be improved in the revised manuscript. We have answered all comments from reviewer and precisely checked all references.
- Introduction
The current key works on the participation of mast cells in the pathogenesis of fibrosis are not cited.
- We thank Reviewer#2 for pointing out the shortcomings of our expression. We have added the description as:
- MCs are also regulators of inflammatory disorders and fibrosis occurred in various organs. Associations between MCs recruitment/infiltration and fibrosis have been found in various tissues [5]. Current studies have found that many MCs products, including—but not limited to—tryptase, chymase, histamine, TGF- β1, IL-13, IL-9, CCL2, platelet-derived growth factor (PDGF), glycosaminoglycan and fibroblast growth factor-2 (FGF-2) can promote fibrosis [5]. MCs have long been regarded as the initiators of allergy and inflammation, as well as the promoters of fibrotic diseases, which are pathogenic. (Page 1-2, line 40-47)
- Origin and heterogeneity of MCs
It is necessary to consider the material of such key works as:
Mast Cell Biology at Molecular Level: a Comprehensive Review. Elieh Ali Komi D, Wöhrl S, Bielory L. Clin Rev Allergy Immunol. 2020
Mast cells as sources of cytokines, chemokines, and growth factors. Mukai K, Tsai M, Saito H, Galli SJ. Immunol Rev. 2018 Mar;282(1):121-150.
- We thank Reviewer#2 to indicate the important points regarding this section. According to the reviewer's comment, we have added the content:
- It is generally believed that MCPs in mice are derived from bone marrow. It has also been reported that MCs are developed from the common myeloid progenitor cells (CMPs) [16]. However, Dahlin et al., who demonstrated that MCPs were derived from multipotential progenitor cells (MMPs) rather than CMPs, objected [17]. (Page 2, line 69-73)
- Likewise, MCs also have significant heterogeneity in the secretion of cytokines, chemokines and growth factors based on different species sources, tissue locations, developmental stages and exposure to inflammatory or immune responses, which was classified and elaborated in detail by Mukai et al. in a review [29]. (Page 3, line 116-120)
We have added reference:
- [16] Voehringer, D. Protective and pathological roles of mast cells and basophils. Nat Rev Immunol 2013, 13, 362-375.
- [17] Elieh Ali Komi, D.; Wohrl, S.; Bielory, L. Mast Cell Biology at Molecular Level: a Comprehensive Review. Clin Rev Allergy Immunol 2020, 58, 342-365.
- [29] Mukai, K.; Tsai, M.; Saito, H.; Galli, S.J. Mast cells as sources of cytokines, chemokines, and growth factors. Immunol Rev 2018, 282, 121-150.
- Pro-allergic and inflammatory MCs – is an incorrect title of this section, since this type of mast cells is not included in the classification.
- We thank Reviewer#2 for pointing out the error in the title. We have rewritten the title as “Pro-allergic and inflammatory actions of MCs” (Page 3, line 131)
- 2. Fibrogenic MCs
What are Fibrogenic MCs? Can the author provide evidence of the existence of a MC of this type? The section name must be changed!
This section needs to be substantially supplemented and expanded.
This section focuses primarily on the mediated profibrotic effects of mast cells, albeit in insufficient detail. Some of the material is outdated works, and, despite their scientific value, should be supplemented by more recent literary sources. In addition, the authors do not consider at all the issues of the effects of the secretome of mast cells on the formation of collagen fibers, and even more so the direct effects on the molecular mechanisms of fibrillogenesis. At the same time, there are serious developments and studies on this matter, published, in particular, in the article:
Atiakshin D, Buchwalow I, Tiemann M. Mast cells and collagen fibrillogenesis. Histochem Cell Biol. 2020 Jul; 154 (1): 21-40.
Data from this study should be included in the review.
- We thank Reviewer#2 to point out the error in the title and indicate the important points regarding the fibrosis. According to the reviewer's comment, we have rewritten the title as “Fibrogenic actions of MCs” (Page 6, line 285) and we have added the content:
- In the process of fiber formation, fiber molecular subunits are first synthesized in fibroblasts. They transfer from the endoplasmic reticulum to the Golgi complex and finally secreted into the intercellular matrix. In the intercellular matrix, the propeptides of procollagen molecule are cleaved by specific proteolytic enzymes and transformed into tropocollagen. After that, the monomeric subunits are assembled and grow linearly and laterally to form the collagen macromolecular complexes that form the fiber, and finally, the fiber is formed [76]. (Page 7, line 294-300)
- In addition, toluidine blue staining showed that MCs was mostly adjacent to fibroblasts in animal skin simulating trauma and it has been found that a variety of cell surface proteins on fibroblasts can connect to the interaction between fibroblasts and MCs, such as membrane-bound stem cell factor, hyaluronic acid receptors, fibrinogen and gap-junctional intercellular communication (GJIC) [76, 81]. (Page 7, line 303-308)
- Current studies have found that many MCs products, including—but not limited to—tryptase, chymase, histamine, TGF- β1, IL-13, IL-9, CCL2, PDGF, glycosaminoglycan and FGF-2 can promote fibrosis [5]. These secretory components of MCs can not only promote fibroblasts to produce collagen but also participate in the extracellular stage of fiber formation [76]. (Page 7, line 309-313)
- Chymase is produced by the degranulation of MCs and is closely related to fibrosis. Chymase promotes fibroblast mitosis and promotes the synthesis and secretion of type I and III collagen in the ECM [82]. (Page 7, line 314-316)
- Chymase also activates endothelin-1, which is thought to promote organ fibrosis [84]. Furthermore, chymase itself can enzymatically cleave the precursors of pro-gelatinase B (MMP-9), TGF-β1 and c-propeptide from type I collagen molecules, which will enhance their activity [76, 85] (Page 7, line 321-324)
- Furthermore, tryptase can not only activate the mitosis and increase the migration activity of fibroblasts but also make them produce type I collagen fibronectin and laminin, which promote the formation of fibrosis [76, 90]. (Page 7, line 335-337)
- In addition, TGF-β1 can enhance fibronectin internalization [96]. (Page 8, line 354-355)
- Glycosaminoglycans secreted by MCs assist fiber formation in the extracellular stage [76]. As mentioned above, in the ECM, the polymerization of tropocollagen macromolecules can be polymerized into microfibers, fibrils and fibers. Glycosaminoglycan can absorb water before that, which helps increasing the concentration at the time of the polymerization, thus facilitating the polymerization of the tropocollagen. Along with this, there is an electrostatic interaction between glycosaminoglycan and collagen. A recent study showed that heparin forms a bridge between two collagen molecules, which makes it possible to regulate the distance between them, thus determining the thickness of the fibrils [101]. Accordingly, MCs, as the only source of heparin and other glycosaminoglycan in tissues, have a great contribution to the formation of fibers. What's more, granules released by MCs can also act as nucleators, which can be used as the starting molecular loci of collagen molecular polymerization [76]. (Page 8, line 363-374)
- In addition, MCs can secrete matrix metalloproteinases (MMPs), an enzyme required for collagen fiber degradation [76]. (Page 8, line 380-381)
We have rechecked all references and have changed or deleted the outdated or inappropriate references:
(All references deleted from this review will be listed at the end of this document)
- Cunningham, M.F.; Docherty, N.G.; Burke, J.P.; O'Connell, P.R. S100A4 expression is increased in stricture fibroblasts from patients with fibrostenosing Crohn's disease and promotes intestinal fibroblast migration. Am J Physiol Gastrointest Liver Physiol 2010, 299, G457-466.
- Johnson, L.A.; Rodansky, E.S.; Sauder, K.L.; Horowitz, J.C.; Mih, J.D.; Tschumperlin, D.J.; Higgins, P.D. Matrix stiffness corresponding to strictured bowel induces a fibrogenic response in human colonic fibroblasts. Inflamm Bowel Dis 2013, 19, 891-903.
- Hinz, B. The myofibroblast: paradigm for a mechanically active cell. J Biomech 2010, 43, 146-155.
- Hinz, B. Formation and function of the myofibroblast during tissue repair. J Invest Dermatol 2007, 127, 526-537.
- Hinz, B.; Phan, S.H.; Thannickal, V.J.; Galli, A.; Bochaton-Piallat, M.L.; Gabbiani, G. The myofibroblast: one function, multiple origins. Am J Pathol 2007, 170, 1807-1816.
- Powell, D.W.; Adegboyega, P.A.; Di Mari, J.F.; Mifflin, R.C. Epithelial cells and their neighbors I. Role of intestinal myofibroblasts in development, repair, and cancer. Am J Physiol Gastrointest Liver Physiol 2005, 289, G2-7.
- Garbuzenko, E.; Nagler, A.; Pickholtz, D.; Gillery, P.; Reich, R.; Maquart, F.X.; Levi-Schaffer, F. Human mast cells stimulate fibroblast proliferation, collagen synthesis and lattice contraction: a direct role for mast cells in skin fibrosis. Clin Exp Allergy 2002, 32, 237-246.
- Brian, W.R.; Sari, M.A.; Iwasaki, M.; Shimada, T.; Kaminsky, L.S.; Guengerich, F.P. Catalytic activities of human liver cytochrome P-450 IIIA4 expressed in Saccharomyces cerevisiae. Biochemistry 1990, 29, 11280-11292.
- Takai, S. Miyazaki, M. A novel therapeutic strategy against vascular disorders with chymase inhibitor. Curr Vasc Pharmacol 2003, 1, 217-224.
- Hatamochi, A.; Fujiwara, K.; Ueki, H. Effects of histamine on collagen synthesis by cultured fibroblasts derived from guinea pig skin. Arch Dermatol Res 1985, 277, 60-64.
- Levi-Schaffer, F. Rubinchik, E. Activated mast cells are fibrogenic for 3T3 fibroblasts. J Invest Dermatol 1995, 104, 999-1003.
- Gailit, J.; Marchese, M.J.; Kew, R.R.; Gruber, B.L. The differentiation and function of myofibroblasts is regulated by mast cell mediators. J Invest Dermatol 2001, 117, 1113-1119.
- Saito, A.; Okazaki, H.; Sugawara, I.; Yamamoto, K.; Takizawa, H. Potential action of IL-4 and IL-13 as fibrogenic factors on lung fibroblasts in vitro. Int Arch Allergy Immunol 2003, 132, 168-176.
- Longley, B.J.; Tyrrell, L.; Ma, Y.; Williams, D.A.; Halaban, R.; Langley, K.; Lu, H.S.; Schechter, N.M. Chymase cleavage of stem cell factor yields a bioactive, soluble product. Proc Natl Acad Sci U S A 1997, 94, 9017-9021.
We have added some new reference:
- [76] Atiakshin, D.; Buchwalow, I.; Tiemann, M. Mast cells and collagen fibrillogenesis. Histochem Cell Biol 2020, 154, 21-40.
- [82] Chen, H.; Xu, Y.; Yang, G.; Zhang, Q.; Huang, X.; Yu, L.; Dong, X. Mast cell chymase promotes hypertrophic scar fibroblast proliferation and collagen synthesis by activating TGF-beta1/Smads signaling pathway. Exp Ther Med 2017, 14, 4438-4442.
- [84] Jing, J.; Dou, T.T.; Yang, J.Q.; Chen, X.B.; Cao, H.L.; Min, M.; Cai, S.Q.; Zheng, M.; Man, X.Y. Role of endothelin-1 in the skin fibrosis of systemic sclerosis. Eur Cytokine Netw 2015, 26, 10-14.
- [90] Blank, U.; Madera-Salcedo, I.K.; Danelli, L.; Claver, J.; Tiwari, N.; Sanchez-Miranda, E.; Vazquez-Victorio, G.; Ramirez-Valadez, K.A.; Macias-Silva, M.; Gonzalez-Espinosa, C. Vesicular trafficking and signaling for cytokine and chemokine secretion in mast cells. Front Immunol 2014, 5, 453.
- [96] Brown, M. O'Reilly, S. The immunopathogenesis of fibrosis in systemic sclerosis. Clin Exp Immunol 2019, 195, 310-321.
- [101] Kulke, M.; Geist, N.; Friedrichs, W.; Langel, W. Molecular dynamics simulations on networks of heparin and collagen. Proteins 2017, 85, 1119-1130.
- Regulatory-type MCs in allergy and inflammation
Regulatory-type MCs - it's not entirely clear what this is about. Maybe about a separate type of mast cells that have regulatory properties? Each mast cell has a wide regulatory spectrum in relation to immunocompetent cells, stromal cells, as well as fibrous and amorphous components of the extracellular matrix. In the opinion of the reviewer, the author should rename "Regulatory-type MCs" in the title of this section of the work; there is no clear evidence of the existence of this special type of mast cells.
- We thank Reviewer#2 for pointing out the error in the title. We have rewritten the title as “Regulatory-type actions of MCs in allergy and inflammation” (Page 8, line 384)
- 1. Regulation of chronic inflammation / fibrosis / wound healing
Why do the authors in this section return to the fibrosis that was discussed earlier?
In addition, the authors cite data from rather old works on the participation of mast cells in regeneration and wound healing. Authors need to add materials from the following key works:
A Review of the Contribution of Mast Cells in Wound Healing: Involved Molecular and Cellular Mechanisms. Komi DEA, Khomtchouk K, Santa Maria PL. Clin Rev Allergy Immunol. 2020 Jun; 58 (3): 298-312.
A Review of the Evidence for and against a Role for Mast Cells in Cutaneous Scarring and Fibrosis. Wilgus TA, Ud-Din S, Bayat A. Int J Mol Sci. 2020
Mast Cells in Skin Scarring: A Review of Animal and Human Research. Ud-Din S, Wilgus TA, Bayat A. Front Immunol. 2020 Sep 30; 11: 552205.
Mast Cells in Diabetes and Diabetic Wound Healing. Dong J, Chen L, Zhang Y, Jayaswal N, Mezghani I, Zhang W, Veves A. Adv Ther. 2020 Nov; 37 (11): 4519-4537.
- We thank Reviewer#2 for pointing out the shortcomings of our expression. According to the reviewer's comment, we have revised and added some content:
- Under the influence of SCF released by keratinocytes as well as CCL2 (MCP-1) and IL-33, MCs gather at the edge of the wound in the first few days [113]. The TNF secreted by them can enhance the expression of XIIIa factor in dermal dendritic cells and then promote hemostasis and clot formation, which help reduce injury [114]. (Page 9, line 406-410)
- MCs can release a variety of substances that interact with fibroblasts to promote wound healing. As mentioned before, proteases released by MCs can chemotaxis fibroblasts and promote their mitosis [76, 90]. In addition, VEGF, IL-4 and basic fibroblast growth factor (bFGF) derived from MCs can stimulate the proliferation of fibroblasts [115]. Subsequently, fibroblasts synthesize fibers to aid wound healing. (Page 9, line 414-419)
- Besides, MCs can secrete a variety of mediators including VEGF, IL-8, IL-4, nerve growth factor (NGF), FGF-2, PDGF and TGF-β1 to facilitate angiogenesis, fibrin production and re-epithelialization [17, 114, 116]. Vascular degeneration is the main physiological process in the remodeling stage, which results in the transformation of granulation tissue into collagen-rich scar avascular tissue [117]. Although there are some objections [118, 119], the vast majority of studies support the involvement of MCs in the process of scar formation [94, 120]. In addition to mediators, GJIC between MCs and fibroblasts or myofibroblasts in granulation tissue also acts as a bridge of cellular communication, mediating excessive fibrosis [81]. (Page 9, line 420-428)
We have rechecked all references and have changed or deleted the outdated or overlapped references:
(All references deleted from this review will be listed at the end of this document)
- Egozi, E.I.; Ferreira, A.M.; Burns, A.L.; Gamelli, R.L.; Dipietro, L.A. Mast cells modulate the inflammatory but not the proliferative response in healing wounds. Wound Repair Regen 2003, 11, 46-54.
- Iba, Y.; Shibata, A.; Kato, M.; Masukawa, T. Possible involvement of mast cells in collagen remodeling in the late phase of cutaneous wound healing in mice. Int Immunopharmacol 2004, 4, 1873-1880.
- Gallant-Behm, C.L.; Hildebrand, K.A.; Hart, D.A. The mast cell stabilizer ketotifen prevents development of excessive skin wound contraction and fibrosis in red Duroc pigs. Wound Repair Regen 2008, 16, 226-233.
- Shiota, N.; Nishikori, Y.; Kakizoe, E.; Shimoura, K.; Niibayashi, T.; Shimbori, C.; Tanaka, T.; Okunishi, H. Pathophysiological role of skin mast cells in wound healing after scald injury: study with mast cell-deficient W/W(V) mice. Int Arch Allergy Immunol 2010, 151, 80-88.
- Dvorak, A.M. Mast cell-derived mediators of enhanced microvascular permeability, vascular permeability factor/vascular endothelial growth factor, histamine, and serotonin, cause leakage of macromolecules through a new endothelial cell permeability organelle, the vesiculo-vacuolar organelle. Chem Immunol Allergy 2005, 85, 185-204.
- Weller, K.; Foitzik, K.; Paus, R.; Syska, W.; Maurer, M. Mast cells are required for normal healing of skin wounds in mice. FASEB J 2006, 20, 2366-2368.
- Takato, H.; Yasui, M.; Ichikawa, Y.; Waseda, Y.; Inuzuka, K.; Nishizawa, Y.; Tagami, A.; Fujimura, M.; Nakao, S. The specific chymase inhibitor TY-51469 suppresses the accumulation of neutrophils in the lung and reduces silica-induced pulmonary fibrosis in mice. Exp Lung Res 2011, 37, 101-108.
- Levi-Schaffer, F. Kupietzky, A. Mast cells enhance migration and proliferation of fibroblasts into an in vitro wound. Experimental Cell Research 1990, 188, 42-49.
We have added some new reference:
- [76] Atiakshin, D.; Buchwalow, I.; Tiemann, M. Mast cells and collagen fibrillogenesis. Histochem Cell Biol 2020, 154, 21-40.
- [90] Blank, U.; Madera-Salcedo, I.K.; Danelli, L.; Claver, J.; Tiwari, N.; Sanchez-Miranda, E.; Vazquez-Victorio, G.; Ramirez-Valadez, K.A.; Macias-Silva, M.; Gonzalez-Espinosa, C. Vesicular trafficking and signaling for cytokine and chemokine secretion in mast cells. Front Immunol 2014, 5, 453.
- [94] Wilgus, T.A.; Ud-Din, S.; Bayat, A. A Review of the Evidence for and against a Role for Mast Cells in Cutaneous Scarring and Fibrosis. Int J Mol Sci 2020, 21, 9673.
- [114] Komi, D.E.A.; Khomtchouk, K.; Santa Maria, P.L. A Review of the Contribution of Mast Cells in Wound Healing: Involved Molecular and Cellular Mechanisms. Clin Rev Allergy Immunol 2020, 58, 298-312.
- [115] Tellechea, A.; Leal, E.C.; Kafanas, A.; Auster, M.E.; Kuchibhotla, S.; Ostrovsky, Y.; Tecilazich, F.; Baltzis, D.; Zheng, Y.; Carvalho, E.; Zabolotny, J.M.; Weng, Z.; Petra, A.; Patel, A.; Panagiotidou, S.; Pradhan-Nabzdyk, L.; Theoharides, T.C.; Veves, A. Mast Cells Regulate Wound Healing in Diabetes. Diabetes 2016, 65, 2006-2019.
- [116] Dong, J.; Chen, L.; Zhang, Y.; Jayaswal, N.; Mezghani, I.; Zhang, W.; Veves, A. Mast Cells in Diabetes and Diabetic Wound Healing. Adv Ther 2020, 37, 4519-4537.
- [117] Nishikori, Y.; Shiota, N.; Okunishi, H. The role of mast cells in cutaneous wound healing in streptozotocin-induced diabetic mice. Arch Dermatol Res 2014, 306, 823-835.
- [119] Willenborg, S.; Eckes, B.; Brinckmann, J.; Krieg, T.; Waisman, A.; Hartmann, K.; Roers, A.; Eming, S.A. Genetic ablation of mast cells redefines the role of mast cells in skin wound healing and bleomycin-induced fibrosis. J Invest Dermatol 2014, 134, 2005-2015.
- [120] Ud-Din, S.; Wilgus, T.A.; Bayat, A. Mast Cells in Skin Scarring: A Review of Animal and Human Research. Front Immunol 2020, 11, 552205.
- In addition, the authors in this section practically ignore the involvement of specific mast cell proteases in the development of inflammatory reactions. Therefore, the role of mast cells in inflammation is not adequately explained. It is necessary to add citations with data on the biological effects of tryptase, chymase, and carboxypeptidase A3, in particular:
Protease profile of normal and neoplastic mast cells in the human bone marrow with special emphasis on systemic mastocytosis. Atiakshin D, Buchwalow I, Horny P, Tiemann M. Histochem Cell Biol. 2021 May; 155 (5): 561-580. "
Mast cell chymase: morphofunctional characteristics. Atiakshin D, Buchwalow I, Tiemann M. Histochem Cell Biol. 2019 Oct; 152 (4): 253-269.
Tryptase as a polyfunctional component of mast cells. Atiakshin D, Buchwalow I, Samoilova V, Tiemann M. Histochem Cell Biol. 2018 May; 149 (5): 461-477.
Localization of Flt1 and tryptase of mast cells in skin wound of rats with type I diabetes: Initial studies. Bayat M, Chien S, Chehelcheraghi F. Acta Histochem. 2021.
- We thank Reviewer#2 to indicate the important points regarding specific mast cell proteases. According to the reviewer's comment, we have revised and added some content:
- MCs have long been regarded as the initiators of immunity and inflammation,which is pathogenic. For example, specific MCs proteases can promote inflammation, such as tryptase, chymase and carboxypeptidase A3 [103-106]. Tryptase was found to promote M1 macrophages polarization and inflammation via PAR2/FOXO1 pathway, which may be related to macrophage‐associated inflammation in obese adipose tissue and atherosclerotic plaque [107]. Tryptase was also found to promote inflammation in osteoarthritis and skin [106, 108]. Similarly, the pro-inflammatory effects of Chymase have been detected in several inflammatory diseases. Studies have shown that the pro-inflammatory effects of chymase may be related to the activation of several cytokines and growth factors, such as IL-1β, IL-6, IL-8, IL-18, TGF-β1, endothelin-1 and -2, and neutrophil-activating peptide 2 (NAP-2) [104]. In addition, the pro-inflammatory effects of carboxypeptidase A3 have also been mentioned in some cases of respiratory inflammation [109-111]. (Page 8, line 385-397)
We have added reference:
- [103] Atiakshin, D.; Buchwalow, I.; Horny, P.; Tiemann, M. Protease profile of normal and neoplastic mast cells in the human bone marrow with special emphasis on systemic mastocytosis. Histochem Cell Biol 2021, 155, 561-580.
- [104] Atiakshin, D.; Buchwalow, I.; Tiemann, M. Mast cell chymase: morphofunctional characteristics. Histochem Cell Biol 2019, 152, 253-269.
- [105] Atiakshin, D.; Buchwalow, I.; Samoilova, V.; Tiemann, M. Tryptase as a polyfunctional component of mast cells. Histochem Cell Biol 2018, 149, 461-477.
- [106] Bayat, M.; Chien, S.; Chehelcheraghi, F. Co- localization of Flt1 and tryptase of mast cells in skin wound of rats with type I diabetes: Initial studies. Acta Histochem 2021, 123, 151680.
- [107] Chen, L.; Gao, B.; Zhang, Y.; Lu, H.; Li, X.; Pan, L.; Yin, L.; Zhi, X. PAR2 promotes M1 macrophage polarization and inflammation via FOXO1 pathway. J Cell Biochem 2019, 120, 9799-9809.
- [108] Wang, Q.; Lepus, C.M.; Raghu, H.; Reber, L.L.; Tsai, M.M.; Wong, H.H.; von Kaeppler, E.; Lingampalli, N.; Bloom, M.S.; Hu, N.; Elliott, E.E.; Oliviero, F.; Punzi, L.; Giori, N.J.; Goodman, S.B.; Chu, C.R.; Sokolove, J.; Fukuoka, Y.; Schwartz, L.B.; Galli, S.J.; Robinson, W.H. IgE-mediated mast cell activation promotes inflammation and cartilage destruction in osteoarthritis. Elife 2019, 8.
- [109] Pejler, G. The emerging role of mast cell proteases in asthma. Eur Respir J 2019, 54.
- [110] Lai, Y.; Altemeier, W.A.; Vandree, J.; Piliponsky, A.M.; Johnson, B.; Appel, C.L.; Frevert, C.W.; Hyde, D.M.; Ziegler, S.F.; Smith, D.E.; Henderson, W.R., Jr.; Gelb, M.H.; Hallstrand, T.S. Increased density of intraepithelial mast cells in patients with exercise-induced bronchoconstriction regulated through epithelially derived thymic stromal lymphopoietin and IL-33. J Allergy Clin Immunol 2014, 133, 1448-1455.
- [111] Takabayashi, T.; Kato, A.; Peters, A.T.; Suh, L.A.; Carter, R.; Norton, J.; Grammer, L.C.; Tan, B.K.; Chandra, R.K.; Conley, D.B.; Kern, R.C.; Fujieda, S.; Schleimer, R.P. Glandular mast cells with distinct phenotype are highly elevated in chronic rhinosinusitis with nasal polyps. J Allergy Clin Immunol 2012, 130, 410-420 e415.
Here are all outdated or overlapping references and deleted in the revised manuscript:
- Bao, Y.; Wang, S.; Gao, Y.; Zhang, W.; Jin, H.; Yang, Y.; Li, J. MicroRNA-126 accelerates IgE-mediated mast cell degranulation associated with the PI3K/Akt signaling pathway by promoting Ca(2+) influx. Exp Ther Med 2018, 16, 2763-2769.
- Beavitt, S.J.; Harder, K.W.; Kemp, J.M.; Jones, J.; Quilici, C.; Casagranda, F.; Lam, E.; Turner, D.; Brennan, S.; Sly, P.D.; Tarlinton, D.M.; Anderson, G.P.; Hibbs, M.L. Lyn-deficient mice develop severe, persistent asthma: Lyn is a critical negative regulator of Th2 immunity. J Immunol 2005, 175, 1867-1875.
- Carlos, D.; Yaochite, J.N.; Rocha, F.A.; Toso, V.D.; Malmegrim, K.C.; Ramos, S.G.; Jamur, M.C.; Oliver, C.; Camara, N.O.; Andrade, M.V.; Cunha, F.Q.; Silva, J.S. Mast cells control insulitis and increase Treg cells to confer protection against STZ-induced type 1 diabetes in mice. Eur J Immunol 2015, 45, 2873-2885.
- Cildir, G.; Pant, H.; Lopez, A.F.; Tergaonkar, V. The transcriptional program, functional heterogeneity, and clinical targeting of mast cells. J Exp Med 2017, 214, 2491-2506.
- Cunningham, M.F.; Docherty, N.G.; Burke, J.P.; O'Connell, P.R. S100A4 expression is increased in stricture fibroblasts from patients with fibrostenosing Crohn's disease and promotes intestinal fibroblast migration. Am J Physiol Gastrointest Liver Physiol 2010, 299, G457-466.
- Dahlin, J.S.; Ding, Z.; Hallgren, J. Distinguishing Mast Cell Progenitors from Mature Mast Cells in Mice. Stem Cells Dev 2015, 24, 1703-1711.
- Di, S.; Ziyou, Y.; Liu, N.F. Pathological Changes of Lymphedematous Skin: Increased Mast Cells, Related Proteases, and Activated Transforming Growth Factor-beta1. Lymphat Res Biol 2016, 14, 162-171.
- Dvorak, A.M. Mast cell-derived mediators of enhanced microvascular permeability, vascular permeability factor/vascular endothelial growth factor, histamine, and serotonin, cause leakage of macromolecules through a new endothelial cell permeability organelle, the vesiculo-vacuolar organelle. Chem Immunol Allergy 2005, 85, 185-204.
- Eckl-Dorna, J.; Pree, I.; Reisinger, J.; Marth, K.; Chen, K.W.; Vrtala, S.; Spitzauer, S.; Valenta, R.; Niederberger, V. The majority of allergen-specific IgE in the blood of allergic patients does not originate from blood-derived B cells or plasma cells. Clin Exp Allergy 2012, 42, 1347-1355.
- Egozi, E.I.; Ferreira, A.M.; Burns, A.L.; Gamelli, R.L.; Dipietro, L.A. Mast cells modulate the inflammatory but not the proliferative response in healing wounds. Wound Repair Regen 2003, 11, 46-54.
- Ehrlich, H.P. A Snapshot of Direct Cell-Cell Communications in Wound Healing and Scarring. Adv Wound Care (New Rochelle) 2013, 2, 113-121.
- Gaca, M.D.; Pickering, J.A.; Arthur, M.J.; Benyon, R.C. Human and rat hepatic stellate cells produce stem cell factor: a possible mechanism for mast cell recruitment in liver fibrosis. J Hepatol 1999, 30, 850-858.
- Gailit, J.; Marchese, M.J.; Kew, R.R.; Gruber, B.L. The differentiation and function of myofibroblasts is regulated by mast cell mediators. J Invest Dermatol 2001, 117, 1113-1119.
- Gallant-Behm, C.L.; Hildebrand, K.A.; Hart, D.A. The mast cell stabilizer ketotifen prevents development of excessive skin wound contraction and fibrosis in red Duroc pigs. Wound Repair Regen 2008, 16, 226-233.
- Gao, Z.G. Jacobson, K.A. Purinergic Signaling in Mast Cell Degranulation and Asthma. Front Pharmacol 2017, 8, 947.
- Garbuzenko, E.; Nagler, A.; Pickholtz, D.; Gillery, P.; Reich, R.; Maquart, F.X.; Levi-Schaffer, F. Human mast cells stimulate fibroblast proliferation, collagen synthesis and lattice contraction: a direct role for mast cells in skin fibrosis. Clin Exp Allergy 2002, 32, 237-246.
- Gaudenzio, N.; Sibilano, R.; Marichal, T.; Starkl, P.; Reber, L.L.; Cenac, N.; McNeil, B.D.; Dong, X.; Hernandez, J.D.; Sagi-Eisenberg, R.; Hammel, I.; Roers, A.; Valitutti, S.; Tsai, M.; Espinosa, E.; Galli, S.J. Different activation signals induce distinct mast cell degranulation strategies. J Clin Invest 2016, 126, 3981-3998.
- Halai, R.; Croker, D.E.; Suen, J.Y.; Fairlie, D.P.; Cooper, M.A. A Comparative Study of Impedance versus Optical Label-Free Systems Relative to Labelled Assays in a Predominantly Gi Coupled GPCR (C5aR) Signalling. Biosensors (Basel) 2012, 2, 273-290.
- Hinz, B. Formation and function of the myofibroblast during tissue repair. J Invest Dermatol 2007, 127, 526-537.
- Hinz, B. The myofibroblast: paradigm for a mechanically active cell. J Biomech 2010, 43, 146-155.
- Hinz, B.; Phan, S.H.; Thannickal, V.J.; Galli, A.; Bochaton-Piallat, M.L.; Gabbiani, G. The myofibroblast: one function, multiple origins. Am J Pathol 2007, 170, 1807-1816.
- Iba, Y.; Shibata, A.; Kato, M.; Masukawa, T. Possible involvement of mast cells in collagen remodeling in the late phase of cutaneous wound healing in mice. Int Immunopharmacol 2004, 4, 1873-1880.
- Idzko, M.; Hammad, H.; van Nimwegen, M.; Kool, M.; Willart, M.A.; Muskens, F.; Hoogsteden, H.C.; Luttmann, W.; Ferrari, D.; Di Virgilio, F.; Virchow, J.C., Jr.; Lambrecht, B.N. Extracellular ATP triggers and maintains asthmatic airway inflammation by activating dendritic cells. Nat Med 2007, 13, 913-919.
- Irani, A.A.; Schechter, N.M.; Craig, S.S.; DeBlois, G.; Schwartz, L.B. Two types of human mast cells that have distinct neutral protease compositions. Proc Natl Acad Sci U S A 1986, 83, 4464-4468.
- Johnson, L.A.; Rodansky, E.S.; Sauder, K.L.; Horowitz, J.C.; Mih, J.D.; Tschumperlin, D.J.; Higgins, P.D. Matrix stiffness corresponding to strictured bowel induces a fibrogenic response in human colonic fibroblasts. Inflamm Bowel Dis 2013, 19, 891-903.
- Junger, W.G. Immune cell regulation by autocrine purinergic signalling. Nat Rev Immunol 2011, 11, 201-212.
- Katz, H.R.; Stevens, R.L.; Austen, K.F. Heterogeneity of mammalian mast cells differentiated in vivo and in vitro. J Allergy Clin Immunol 1985, 76, 250-259.
- Kawakami, T. Galli, S.J. Regulation of mast-cell and basophil function and survival by IgE. Nat Rev Immunol 2002, 2, 773-786.
- Krystel-Whittemore, M.; Dileepan, K.N.; Wood, J.G. Mast Cell: A Multi-Functional Master Cell. Front Immunol 2015, 6, 620.
- Kyritsi, K.; Kennedy, L.; Meadows, V.; Hargrove, L.; Demieville, J.; Pham, L.; Sybenga, A.; Kundu, D.; Cerritos, K.; Meng, F.; Alpini, G.; Francis, H. Mast Cells Induce Ductular Reaction Mimicking Liver Injury in Mice Through Mast Cell-Derived Transforming Growth Factor Beta 1 Signaling. Hepatology 2020.
- Levi-Schaffer, F. Kupietzky, A. Mast cells enhance migration and proliferation of fibroblasts into an in vitro wound. Experimental Cell Research 1990, 188, 42-49.
- Li, Z.; Liu, S.; Xu, J.; Zhang, X.; Han, D.; Liu, J.; Xia, M.; Yi, L.; Shen, Q.; Xu, S.; Lu, L.; Cao, X. Adult Connective Tissue-Resident Mast Cells Originate from Late Erythro-Myeloid Progenitors. Immunity 2018, 49, 640-653 e645.
- Longley, B.J.; Tyrrell, L.; Ma, Y.; Williams, D.A.; Halaban, R.; Langley, K.; Lu, H.S.; Schechter, N.M. Chymase cleavage of stem cell factor yields a bioactive, soluble product. Proc Natl Acad Sci U S A 1997, 94, 9017-9021.
- Lu, J.; Chen, B.; Li, S.; Sun, Q. Tryptase inhibitor APC 366 prevents hepatic fibrosis by inhibiting collagen synthesis induced by tryptase/protease-activated receptor 2 interactions in hepatic stellate cells. Int Immunopharmacol 2014, 20, 352-357.
- MacGlashan, D., Jr. Undem, B.J. Inducing an anergic state in mast cells and basophils without secretion. J Allergy Clin Immunol 2008, 121, 1500-1506, 1506 e1501-1504.
- Meininger, J.; Yano, H.; Rottapel, R.; Bernstein, A.; Zsebo, K.M.; Zetter, B.R. The c-kit receptor ligand functions as a mast cell chemoattractant. Blood 1992, 79, 958-963.
- Meixiong, J. Dong, X. Mas-Related G Protein-Coupled Receptors and the Biology of Itch Sensation. Annu Rev Genet 2017, 51, 103-121.
- Metcalfe, D.D.; Baram, D.; Mekori, Y.A. Mast cells. Physiol Rev 1997, 77, 1033-1079.
- Nam, S.T.; Park, Y.H.; Kim, H.W.; Kim, H.S.; Lee, D.; Lee, M.B.; Kim, Y.M.; Choi, W.S. Suppression of IgE-mediated mast cell activation and mouse anaphylaxis via inhibition of Syk activation by 8-formyl-7-hydroxy-4-methylcoumarin, 4mu8C. Toxicol Appl Pharmacol 2017, 332, 25-31.
- Nilsson, G.; Butterfield, J.H.; Nilsson, K.; Sa, S. Stem cell factor is a chemotactic factor for human mast cells. Journal of Immunology 1994, 153, 3717-3723.
- Olivera, A.; Beaven, M.A.; Metcalfe, D.D. Mast cells signal their importance in health and disease. J Allergy Clin Immunol 2018, 142, 381-393.
- Patou, J.; Holtappels, G.; Affleck, K.; van Cauwenberge, P.; Bachert, C. Syk-kinase inhibition prevents mast cell activation in nasal polyps. Rhinology 2011, 49, 100-106.
- Powell, D.W.; Adegboyega, P.A.; Di Mari, J.F.; Mifflin, R.C. Epithelial cells and their neighbors I. Role of intestinal myofibroblasts in development, repair, and cancer. Am J Physiol Gastrointest Liver Physiol 2005, 289, G2-7.
- Rak, G.D.; Osborne, L.C.; Siracusa, M.C.; Kim, B.S.; Wang, K.; Bayat, A.; Artis, D.; Volk, S.W. IL-33-Dependent Group 2 Innate Lymphoid Cells Promote Cutaneous Wound Healing. J Invest Dermatol 2016, 136, 487-496.
- Ribatti, D. Crivellato, E. Mast cell ontogeny: an historical overview. Immunol Lett 2014, 159, 11-14.
- Sakaguchi, S.; Vignali, D.A.; Rudensky, A.Y.; Niec, R.E.; Waldmann, H. The plasticity and stability of regulatory T cells. Nat Rev Immunol 2013, 13, 461-467.
- Shiota, N.; Nishikori, Y.; Kakizoe, E.; Shimoura, K.; Niibayashi, T.; Shimbori, C.; Tanaka, T.; Okunishi, H. Pathophysiological role of skin mast cells in wound healing after scald injury: study with mast cell-deficient W/W(V) mice. Int Arch Allergy Immunol 2010, 151, 80-88.
- Sonoda, T.; Hayashi, C.; Kitamura, Y. Presence of mast cell precursors in the yolk sac of mice. Dev Biol 1983, 97, 89-94.
- Talay, O.; Yan, D.; Brightbill, H.D.; Straney, E.E.; Zhou, M.; Ladi, E.; Lee, W.P.; Egen, J.G.; Austin, C.D.; Xu, M.; Wu, L.C. Addendum: IgE+ memory B cells and plasma cells generated through a germinal-center pathway. Nat Immunol 2013, 14, 1302-1304.
- Tkaczyk, C.; Iwaki, S.; Metcalfe, D.D.; Gilfillan, A.M. Roles of adaptor molecules in mast cell activation. Chem Immunol Allergy 2005, 87, 43-58.
- Valent, P.; Akin, C.; Bonadonna, P.; Hartmann, K.; Brockow, K.; Niedoszytko, M.; Nedoszytko, B.; Siebenhaar, F.; Sperr, W.R.; Oude Elberink, J.N.G.; Butterfield, J.H.; Alvarez-Twose, I.; Sotlar, K.; Reiter, A.; Kluin-Nelemans, H.C.; Hermine, O.; Gotlib, J.; Broesby-Olsen, S.; Orfao, A.; Horny, H.P.; Triggiani, M.; Arock, M.; Schwartz, L.B.; Metcalfe, D.D. Proposed Diagnostic Algorithm for Patients with Suspected Mast Cell Activation Syndrome. J Allergy Clin Immunol Pract 2019, 7, 1125-1133 e1121.
- Valent, P.; Akin, C.; Hartmann, K.; Nilsson, G.; Reiter, A.; Hermine, O.; Sotlar, K.; Sperr, W.R.; Escribano, L.; George, T.I.; Kluin-Nelemans, H.C.; Ustun, C.; Triggiani, M.; Brockow, K.; Gotlib, J.; Orfao, A.; Schwartz, L.B.; Broesby-Olsen, S.; Bindslev-Jensen, C.; Kovanen, P.T.; Galli, S.J.; Austen, K.F.; Arber, D.A.; Horny, H.P.; Arock, M.; Metcalfe, D.D. Advances in the Classification and Treatment of Mastocytosis: Current Status and Outlook toward the Future. Cancer Res 2017, 77, 1261-1270.
- Wang, L.; Sikora, J.; Hu, L.; Shen, X.; Grygorczyk, R.; Schwarz, W. ATP release from mast cells by physical stimulation: a putative early step in activation of acupuncture points. Evid Based Complement Alternat Med 2013, 2013, 350949.
- Welle, M. Development, significance, and heterogeneity of mast cells with particular regard to the mast cell-specific proteases chymase and tryptase. J Leukoc Biol 1997, 61, 233-245.
- Weller, K.; Foitzik, K.; Paus, R.; Syska, W.; Maurer, M. Mast cells are required for normal healing of skin wounds in mice. FASEB J 2006, 20, 2366-2368.
- Wulff, B.C.; Parent, A.E.; Meleski, M.A.; DiPietro, L.A.; Schrementi, M.E.; Wilgus, T.A. Mast cells contribute to scar formation during fetal wound healing. J Invest Dermatol 2012, 132, 458-465.
- Wygrecka, M.; Dahal, B.K.; Kosanovic, D.; Petersen, F.; Taborski, B.; von Gerlach, S.; Didiasova, M.; Zakrzewicz, D.; Preissner, K.T.; Schermuly, R.T.; Markart, P. Mast cells and fibroblasts work in concert to aggravate pulmonary fibrosis: role of transmembrane SCF and the PAR-2/PKC-alpha/Raf-1/p44/42 signaling pathway. Am J Pathol 2013, 182, 2094-2108.
- Yasuda, K.; Muto, T.; Kawagoe, T.; Matsumoto, M.; Sasaki, Y.; Matsushita, K.; Taki, Y.; Futatsugi-Yumikura, S.; Tsutsui, H.; Ishii, K.J.; Yoshimoto, T.; Akira, S.; Nakanishi, K. Contribution of IL-33-activated type II innate lymphoid cells to pulmonary eosinophilia in intestinal nematode-infected mice. Proc Natl Acad Sci U S A 2012, 109, 3451-3456.
- Yu, M.; Tsai, M.; Tam, S.Y.; Jones, C.; Zehnder, J.; Galli, S.J. Mast cells can promote the development of multiple features of chronic asthma in mice. J Clin Invest 2006, 116, 1633-1641.
- Zimmermann, C.; Troeltzsch, D.; Gimenez-Rivera, V.A.; Galli, S.J.; Metz, M.; Maurer, M.; Siebenhaar, F. Mast cells are critical for controlling the bacterial burden and the healing of infected wounds. Proc Natl Acad Sci U S A 2019, 116, 20500-20504.
Round 2
Reviewer 1 Report
Authors have made the solicited corrections and the review contains interesting information and points of view. However, English style should be improved and some minor revisions must be performed:
Figure 1 shows now the adapter NTAL, but it is written NATL. Please correct.
Line 416 says that MCs can chemotaxis fibroblasts. It should say MCs products can chemoattract fibroblasts. Please correct.
Reviewer 2 Report
This work can be accept in present form